# Short and long-term stratospheric impact of smoke from the 2019/2020 Australian wildfires

Johan Friberg[1], Bengt G. Martinsson[1], and Moa K. Sporre[1]

[1]Department of Physics, Lund University, Lund, 22100, Sweden

*Correspondence to*: johan.friberg@nuclear.lu.se

**Abstract.** In the end of December 2019 and beginning of 2020, massive firestorms in Australia formed pyrocumulonimbus clouds (PyroCb) that acted like enormous smokestacks, pumping smoke to the upper troposphere and stratosphere. We study the smoke with data from four satellite-based sensors: the aerosol observation platforms CALIOP, OMPS-LP, and OMPS-NM, and water vapor retrievals from MLS. Smoke was lofted to the upper troposphere and stratosphere during two events and
spread almost exclusively within the extratropics. Smoke from the 1st event, starting Dec 29, was injected directly into the stratosphere by PyroCbs causing a rapid initial increase in AOD. CALIOP identifies a rapid decline in this stratospheric smoke (half-life: 10 days), not captured in previous studies of the Australian fires, indicating photochemical processing of organic aerosol. This decay rate is in line with model predictions of mid-tropospheric organic aerosol loss by photolytic removal and is in agreement with our estimates of decay rates after the North American fires in Aug 2017. PyroCbs from the 2nd event, Jan
4, injected small amounts of smoke directly into the stratosphere. Large amounts of smoke was injected to the upper troposphere, from where it ascended into the stratosphere during several weeks, forming a second peak in the aerosol load. Hence, we find that PyroCbs can impact the stratospheric aerosol load both via direct injection to the stratosphere, and through injection of smoke to the upper troposphere from where the smoke ascends into the stratosphere. The stratospheric AOD from the 2nd event fires decreased more slowly than the AOD from the 1st event, likely due to a combination of photolytic loss
starting already in the troposphere and continued supply of smoke from the upper troposphere offsetting the loss rate. Together these injections gave a major increase in the aerosol load during almost one year.

## 1 Introduction

The stratospheric aerosol scatters and absorbs solar radiation cooling the Earth's surface. Submicron particles can remain suspended for months or years in the stratosphere owing to their low settling rates, lack of precipitation in the stratosphere,
and the stratospheric transport patterns (Kremser et al., 2016). They follow the air stream and are removed from the stratosphere mainly in the midlatitudes and Polar Regions.

The Brewer-Dobson circulation moves air from the tropics to extratropics, where it descends into the lowermost stratosphere (LMS) and eventually to the upper troposphere. Residence times are years in its deep branch, and months in its shallow branch

and LMS. These three layers hold 1/3 each of the global stratospheric aerosol load in periods of stratospheric background conditions (Andersson et al., 2015; Friberg et al., 2018). Variability in the stratospheric aerosol load is driven mainly by volcanic injections of particle forming $SO_2$ (Kremser et al., 2016; Solomon et al., 2011), but wildfires have been shown to contribute substantially in recent years (Peterson et al., 2021).

Wildfires can form so-called pyrocumulonimbus clouds (PyroCbs) that act as giant smokestacks injecting smoke to high altitudes. This phenomenon was revealed already more than 20 years ago, as some stratospheric aerosol layers could not be connected to volcanism (M. Fromm et al., 2000; M. D. Fromm & Servranckx, 2003). There have been many observations of stratospheric wildfire smoke since then, but its impact on the stratospheric aerosol load and climate has been considered low compared to the volcanic (Kremser et al., 2016; Thomason et al., 2018; Vernier et al., 2011). However, recently two massive
events have turned the attention to wildfire smoke.

Massive fires in western North America in August 2017 formed PyroCb that had a remarkable impact on the stratospheric aerosol load and were the largest occasion in the satellite records (S. M. Khaykin et al., 2018; Peterson et al., 2018). Dense smoke layers rose to more than 23 km altitude due to radiation heating of the black carbon (BC) rich aerosol (Yu et al., 2019).
The smoke properties were investigated (Das et al., 2021; Haarig et al., 2018; Martinsson et al., 2022) and its impact on the stratospheric aerosol load just after the fires was estimated to be similar to recent volcanic eruptions (Peterson et al., 2018).

A similar event was observed during the gigantic fires in Eastern Australia in 2019/2020 (Kablick et al., 2020; S. Khaykin et al., 2020). The Austral spring and summer of 2019/2020 were exceptionally hot and dry, and the fire season started earlier
than normally. It was unprecedented both in number of fires and size, and more than 20% of the Australian temperate forest was lost (Abram et al., 2021; Boer et al., 2020). Smoke spread over most of Eastern Australia. 20 PyroCbs injected smoke to the stratosphere during two events, Dec 29-31 and Jan 4 (Peterson et al., 2021). Light absorption of the BC-containing aerosol resulted in three vortex-like structures in the stratosphere (Kablick et al., 2020; S. Khaykin et al., 2020; Lestrelin et al., 2021; Peterson et al., 2021). Smoke layers were seen deep into the stratosphere in the beginning of January, and one of them rose to
more than 35 km altitude by radiation heating (Khaykin et al., 2020).

Most smoke encounters in the stratosphere have been explained through upward transport by pyrocumulonimbus clouds, but studies in recent years suggest that further transport mechanisms cause cross-tropopause transport of smoke. The North American wildfires in Aug 2017 showed that self-lofting by radiative heating of the dense smoke layers caused smoke to rise
from the tropopause into the LMS (e.g. Khaykin et al., 2018; Peterson et al., 2018). Ohneiser et al. (2021) suggested self-lofting of smoke from the mid-troposphere as cause of extensive aerosol layers in the Arctic stratosphere in the end of 2019 and beginning of 2020. Whether those aerosol layers consisted of sulfate or sulfate-covered smoke particles is under debate (Boone et al., 2022; Knepp et al., 2022). Most recently, (Ohneiser et al., 2023) computed heating and lofting rates for light-

absorbing smoke layers throughout the troposphere and the lower stratosphere. Their studies indicate that smoke layers can rise from the UT to the stratosphere via radiation heating. Also, convection downwind fires and isentropic cross-tropopause transport have been suggested as causes of the large stratospheric smoke amount after the Australian fires covered in our work (Hirsch & Koren, 2021; Magaritz-Ronen & Raveh-Rubin, 2021).

Recent findings show that 80-90% of the stratospheric smoke AOD after the 2017 North American fires was lost in the first few months after injection indicating photochemical processing of organics in the smoke (Martinsson et al., 2022), whereas the remaining aerosol stayed in the stratosphere for a year (Martinsson et al., 2022). We present, for the first time, evidence that this phenomenon occurred also in the Australian wildfire smoke layers. Furthermore, we find that part of the smoke lingered in the troposphere for more than one week while gradually entering the stratosphere.

## 2 Methods

### 2.1 Aerosol data

The smoke was observed using lidar data from the NASA satellite borne CALIOP instrument. We used the latest version of the rawest product available (V4-10, Level 1B). Backscattering coefficients were computed by correcting for light attenuation by air molecules (including ozone absorption), and for particles in the stratosphere. The molecular part was estimated using modelling data of the ozone and air densities provided in the CALIOP files compiled by NASA Langley Research Center (Friberg et al., 2018). The wildfire smoke is optically dense and strongly attenuated the lidar signals. The particle light attenuation was computed from the lidar signals themselves in an iterative approach explained in Martinsson et al. (2022). This procedure retrieves also the extinction-to-backscattering ratio, the so-called lidar ratio, used to compute aerosol extinction coefficients from the CALIOP lidar backscattering data. CALIOP has a polarization filter, separating backscattered light into parallel and perpendicular polarization. The ratio of the two forms the volume depolarization ratios used for ice cloud screening of the entire dataset. The volume depolarization ratio describes the properties of the complete volume of air, i.e. the aerosol particles together with the air. To study the temporal evolution of the smoke particles we therefore compiled also the particle depolarization ratios for individual smoke layers (Martinsson et al., 2022), which describes the properties of the particles themselves.

Ice clouds were removed in the lowest 3 kilometres of the stratosphere. CALIOP data were averaged to 8 km horizontal resolution, and volume depolarization ratios above 0.20 were classified as clouds. The process is described in more detail in Martinsson et al. (2022).

Further aerosol data was compiled from the limb-scatter observing instrument OMPS-LP (Ozone Mapping and Profiler Suite Limb Profiler). We used level 2.0 version 5-10 light extinction data (Suomi-NPP OMPS LP L2 AER Daily Product, version

2.0 (Taha et al., 2021)) of two wavelengths (745 and 997 nm). The OMPS-LP wavelength (510 nm), most similar to the CALIOP wavelength (532 nm), and used in Martinsson et al. (2022), is unfortunately not reliable in the Southern Hemisphere (Taha et al., 2021). Data was filtered to minimize influence of ice-clouds and polar stratospheric clouds, and flags were used to prevent influence from erroneous data. A detailed description on this approach can be found in Martinsson et al. (2022).


Stratospheric AODs were computed by integration of the aerosol extinction coefficients from the tropopause to 35 km altitude, as well as in selected layers of the stratosphere, where the LMS was defined as the layer between the tropopause and the 380 K isentrope. Tropopause heights from the MERRA-2 reanalysis were retrieved from the CALIOP and OMPS-LP files provided by NASA.


The UV aerosol index (UVAI) Level 3 data (V2.1) from OMPS-NM were used to track horizontal smoke transport. The data product is compiled from observations at 379 and 340 nm (Torres, 2019). It indicates presence of UV-absorbing aerosol particles and increases with altitude, making it well suited for tracking BC containing wildfire smoke in the UT and stratosphere. Data were screened based on NASA recommendations on data usage

(https://ozoneaq.gsfc.nasa.gov/docs/NMTO3-L3_Product_Descriptions.pdf). Horizontal UVAI distributions were combined with vertical information from CALIOP to identify smoke transport to the upper troposphere and stratosphere from different fire events.

## 2.2 Identifying smoke from different events

Smoke from the different fire events was identified based on daily maps over the UVAI, stratospheric wind directions, and

altitude distributions from CALIOP curtains. We tracked the motion day to day of the central parts of the fire events (S1), and used the information to classify the individual smoke layers described above. OMPS-NM shows two separate major events of increased ultra-violet (UV) aerosol index from the 2019 – 2020 Australian wildfires, first observed on Dec 29 and Jan 4 as described in Peterson et al. (2021). Some days the two events overlapped horizontally. In such case additional information on altitude from CALIOP was used, because the smoke from the two main events on any given day differed markedly in altitude

(Supplementary). We will elaborate more on this in the Results section.

The depolarization ratios for smoke from the 2nd fire were clearly lower than those for smoke from the 1st fire, as seen in Figures 1 and 2, as well as in the supplementary (Fig S2-49. This difference remains for more than one month, i.e., smoke layers from the 2nd fire continues to have lower depolarization ratios than smoke from the 1st fire. This particle optical property

verifies that we have classified the smoke successfully.

CALIOP provides vertical distributions of aerosol and clouds. Besides the attenuated backscattering provided as curtain plots, information on particle morphology (depolarization ratio, where zero corresponds to spherical particles) and particle size (color

ratio, i.e. ratio in attenuated backscattering of wavelength 1064 nm to 532 nm). Figures 1 and 2 show these three CALIOP features over the regions with increased UVAI from the first days after the 1st and 2nd fire event, respectively. Non-cloud features can be identified by strong backscattering signal in connection with depolarization ratios less than approximately 0.2 and color ratios well below 1. For an example see the observation on 2020-01-03 at 10:07 UTC (Figure 1n-p), where a thin smoke layer in the tropopause region resides over deep cloud layers. In optically thick smoke layers, there is a shift in color ratio from a low value at the top to significantly higher values lower down in the smoke layer. These increased values deep into the layers are artefacts caused by stronger attenuation for the shorter wavelength. The signal from the layers closest to the satellite (at the layer top) is less affected by attenuation, whereas deeper into the layer the shortwave signal (532 nm) becomes attenuated more than the longwave one (1064 nm) (Martinsson et al., 2022).

## 2.3 Water vapor observations

Satellite observations of water vapor from the Microwave limb-sounder (MLS) aboard the Aura satellite was used together with aerosol data. We use the level 2 night-time data version 5.0-1.0a data of individual smoke layers. Data is provided at 12 levels per decade change of pressure for 1000 – 1 hPa. Low altitude data were excluded to reduce impact of the strong gradient in $H_2O$ mixing ratio across the tropopause. The highest peak pressure was 73 hPa (average 38 hPa), which is lower than reported in our recent study Martinsson et al. (2022) due to the higher tropopause altitude caused by the lower latitudes in the present study.

MLS data were used in comparison with CALIOP data. A shift down of CALIPSO orbit in September 2018 to the CloudSat level, caused variable horizontal distance between CALIOP aerosol and MLS $H_2O$ observations. Measurements with horizontal distances less than 330 km (average 180 km) were used, which lead to periodical loss of data. The data were screened using recommendations by the MLS team (Livesey et al., 2022).

## 3 Results and Discussions

### 3.1 Smoke distribution in the stratosphere

The CALIOP satellite instrument observed a dense smoke layer at 11-16 km altitude located around the tropopause over the Tasman Sea (Figure. 1a-d), causing clear increase in the aerosol load in the LMS and shallow BD branch already at New Year's Eve (Figure 4). Large amounts of smoke were observed in the following days. Strong smoke signals were seen spread within the southern midlatitudes stratosphere in the beginning of January (Figure 3c, and Figure 4b-d) and continued to be strong during the rest of the month.

The aerosol load was low in the southern extratropical stratosphere before the smoke injection by the Australian fires (Figure 3a). Volcanic perturbations were present at 20 km altitude in the tropics and in the northern extratropics. These stem from eruptions in June and August 2019 by the volcanic eruptions of Ulawun and Raikoke (Kloss et al., 2020), which had low impact on the southern extratropics. In fact, the stratospheric aerosol load was lower in the southern extratropics than anywhere else.

The smoke spread latitudinally, almost exclusively to the south (Figure 3). Figure 4 illustrates the AOD in three stratospheric layers. Most of the smoke stayed below the 470 K isentrope (the two lowest layers, Figure 4b-c), but a minor part of the smoke rose by radiation heating, to the layer with the deep BD branch (Figure 4a) where it continued to rise. A clear AOD increase was evident in the Southern mid and high latitudes persisting throughout 2020.

## 3.2 Wildfire smoke compared to volcanism

We find that the injected wildfire smoke increased the stratospheric aerosol load by volcanic proportions (Figure 5a). The fires induced more than three times higher AOD increase in its first year than the North American fires in 2017 did, and slightly higher than the Calbuco eruption in 2015 (Table 1). That eruption gave the largest volcanic impact on the Southern Hemisphere since the Mount Pinatubo eruption in 1991 (Friberg et al., 2018; Martinsson et al., 2022; Rieger et al., 2015; Thomason et al., 2018). The impact of the Australian fires was only matched in size by the large eruptions of Sarychev (2009) and Raikoke (2019), which both occurred in the Northern Hemisphere (Table 1).

Stratospheric background aerosol consists mainly of sulfurous and carbonaceous compounds (Martinsson et al., 2019). Volcanic aerosol contains large amounts of sulfurous, carbonaceous and crustal components (Andersson et al., 2013; Friberg et al., 2014; Martinsson et al., 2009), whereas smoke mainly consists of organic compounds and BC (Garofalo et al., 2019). The smoke had a rather different impact on the stratospheric aerosol load than the volcanic particles from the Calbuco eruption had (Figure 5a). The rise in AOD during several months after the volcanic injection stems from prolonged particle formation from volcanic $SO_2$ and particle growth. Conversely, wildfire smoke particles are mixtures of primary BC particles, and organics that form within hours or days, explaining the initial rapid rise in the AOD after the fires. In the CALIOP data, a second peak in AOD arose a few weeks after the first. We will elaborate more on this unexpected feature in the following sections.

The AOD evolution showed similar patterns in the two lower layers (Figure 5b). The first peak in AOD is seen early both in the mid- and lowest layer similar to our observations of smoke from the North American fires (Martinsson et al., 2022). The upper layer shows a slow rise in AOD, due to the time required for smoke to rise from the mid layer. This rise was also observed by the limb-scattering instrument OMPS-LP (S. Khaykin et al., 2020) . However, that AOD increase constitutes only a small portion of the total stratospheric AOD from the fires (Figure 5b).

The rapid rise in AOD after the fires is not seen in the OMPS-LP data (Figure 6). CALIOP data reveal that the smoke had twice the peak increase over the background in AOD as the eruption of Calbucco did. Studies based on OMPS-LP data (S. Khaykin et al., 2020) report almost indistinguishable peak impact on the AOD from these two events. The OMPS-LP AODs after the fires increased much more slowly than for CALIOP and did not capture the first peak in the AOD in Figure 5a. This discrepancy is explained by differences in observation systems. OMPS-LP suffers from event termination already at low light extinction, which inhibits quantification of dense aerosol layers such as fresh wildfire smoke (M. Fromm et al., 2014; Lurton et al., 2018; Martinsson et al., 2022). The line-of-sight is orders of magnitude longer for the limb-viewer OMPS-LP than for the nadir-viewer CALIOP, causing the difficulty of observing dense aerosol layers (Martinsson et al., 2022). Hence, CALIOP suffers less from light attenuation and the data can be corrected for light attenuation from the smoke particles, enabling us to compute the AOD of also the densest smoke layers (Martinsson et al., 2022). After 1 – 2 months the limb viewer problem with event termination is reduced, making a comparison of the different instruments feasible. The evolution in stratospheric AOD for the two instruments are compared in Figure 6 illustrating the slower rise for OMPS-LP. As pointed out in the Methods section, the OMPS-LP wavelength closest to CALIOP is not useful in the southern hemisphere (Taha et al., 2021). The light at longer wavelengths (OMPS-LP) are scattered less than the shorter (CALIOP), resulting in lower AODs for OMPS-LP. Similarly slow rise in the AOD from OMPS-LP was shown for smoke from the 2017 North American fires (Martinsson et al., 2022). Our present study shows another example of when space borne lidar is required for quantification of the stratospheric AOD when dense aerosol layers are present.

### 3.3 Stratospheric smoke from two events

We find that smoke was transported to the stratosphere from two fire events, by tracking smoke back to fires in Eastern Australia, combining CALIOP with OMPS-NM (Figure S1). The 1st event PyroCbs started on December 29 (Peterson et al., 2021), and the first CALIOP observations were two days later, on New Year's Eve. Those injections positioned smoke directly around the tropopause, i.e., partly in the stratosphere (Figure 1a-d). One large dense smoke layer was transported east from the fire region and was stuck for weeks over the Southeastern Pacific (Figure 7a and S1), where it formed a vortex isolating it from mixing with surrounding air (Kablick et al., 2020; S. Khaykin et al., 2020). The isolation made it easier to track this smoke. We find that the large dense smoke layer rose by a mean velocity of 260 m/day during the first 50 days after the PyroCb injections (Figure 7b), similar to previously reported figures (S. Khaykin et al., 2020).

We also identified several other smoke layers from the 1st event located at lower stratospheric altitude and not connected to the large dense smoke layer (Figure S1). Horizontally, these were transported more rapidly and could not be tracked during as many days due to mixing with the surrounding air.

The 2nd fire event occurred on January the 4th, but smoke from this event showed only little immediate stratospheric influence (Figure 2, S8-S16). Also Peterson et al. (2021) reported much larger stratospheric impact from the 1st fire, based on studies of

the fires' immediate impact (2021). 10 days after the PyroCb formations we start to see more stratospheric influence (Figure 7). CALIOP images reveal addition of large dense smoke layers to the upper troposphere after Jan 4 (Figure 2). We studied the temporal evolution of these smoke layer's position relative to the tropopause (Figure 11). The smoke layers are clearly located below the tropopause in the first days after the 2nd fire with minor over-shooting parts (e.g. Figure 2f and j). Over time, more and more smoke appears in the stratosphere. Hence, the smoke was transported gradually across the tropopause in the weeks following the fire injections to the upper troposphere.. We kept following this smoke in the stratosphere for 20 days. Interestingly, it rose at approximately the same rate as the large, isolated smoke layer from the 1st event, 250 m/day (Figure 7b).

### 3.4 Transformation of smoke

By studying individual smoke layers, we find evidence of morphological transformation of the smoke particles during the first month after PyroCb injections. The CALIOP instrument is depolarization sensitive. Non-spherical particles depolarize the scattered light increasing the depolarized signal retrieved by the sensor. We find a steady increase in the particle depolarization ratio in stratospheric smoke from both the 1st and 2nd event (Figure 7c). The trend lasts more than 30 days in the isolated layer from the 1st event after which the particle depolarization ratio becomes stable at a value of 0.15. A similar trend was observed in the weeks following the August 2017 fires in western North America (Martinsson et al., 2022), whereas the opposite trend was observed when comparing fresh smoke with aged smoke well mixed with the background aerosol (Baars et al., 2019). The depolarization ratio of the aerosol from the 2nd event deviates clearly from that of the 1st event by being much lower. We will discuss this difference further in a section below.

### 3.5 Separating data from the two events

To study the individual stratospheric impact of the two events, we need to separate data into two groups. (Peterson et al., 2021) reported that PyroCbs reached the stratosphere mainly during the 1st event fires, and to a lesser extent from the 2nd event. Figure 1 shows that smoke from the 1st event reached the stratosphere shortly after the fires, whereas large amounts of smoke from the 2nd event reached the upper troposphere (Figure 2). We do not see evidence of large direct smoke injection to the stratosphere from the 2nd event fires in the CALIOP data, neither in the nighttime nor in the daytime data. Hence, most of the immediate stratospheric impact stem from the 1st event.

Smoke from the 1st event rose markedly in the stratosphere before smoke from the 2nd event entered the stratosphere (Figure 7b and 8a-c). Also their depolarization ratios differed markedly (Figure 7c and supplementary). Clear differences between the 1st and 2nd injection events are evident in the time-altitude distributions (Figure 8) of the extinction coefficients, scattering ratios, and depolarization ratios, remaining over the course of more than two months. These parameters all show rising smoke in the weeks after the 1st event with particle depolarization ratios that increases over time (Figure 8c). In mid-January, smoke with lower depolarization ratios started to ascend in the stratosphere (Figure 7c), connecting the smoke below the minimum in

Figure 8c to the 2nd event that ascended later into the stratosphere. The minima in Figure 8, occurring in between the smoke occurrences from the two injection events illustrates the impact from a rapid stratospheric injection from mainly the 1st fire event, and slow transport of smoke to the stratosphere from the 2nd event. We use this minimum to separate smoke data from the two fires (dashed lines in Figure 8) to form the AOD of the two events and investigate their individual impact on the stratospheric AOD.

### 3.6 Evolution of the smoke

The smoke AOD from the 1st event decreased rapidly over the first weeks, followed by slow decrease until spring (Figure 9a). Similarly rapid decline in smoke AOD and increasing particle depolarization ratio was seen for stratospheric smoke in our earlier study on the western North American wildfires in August 2017 (Martinsson et al., 2022). We have considered transport out of the stratosphere, sedimentation, cloud formation, and hygroscopic growth/shrinkage as explanations of the decline and found that loss of material from the particles by photolysis is the plausible explanation for the decline (Martinsson et al., 2022). The long residence time due to the practically absent wet deposition in the stratosphere makes the effects of photolysis simpler to study compared with the troposphere. The importance of photolysis as a removal mechanism of organic aerosol is also supported by studies of photolysis in numerous laboratory experiments (Molina et al., 2004; Sareen et al., 2013) and by modeling (Hodzic et al., 2015; Zawadowicz et al., 2020).

The trend of decreasing stratospheric AOD after the 1st fire event together with increasing particle depolarization ratio over time suggests that photolytic loss depletes organic aerosol in the smoke. Thus, the BC fraction in the smoke will increase over time and may eventually constitute most of the smoke particles mass. We therefore interpret the morphological transformation (depolarization ratio) and AOD decrease after the 1st event as decay of organic aerosol in the stratosphere.

The particle depolarization ratio is much lower for smoke from the 2nd event than from the 1st (Figure 1, 2, 7c, and 8c). This difference may be found in the history of these smoke layers. Depolarization ratios for tropospheric smoke is lower than stratospheric (Haarig et al., 2018). The low depolarization ratios for smoke from the 2nd event indicate (chemical) processing of the smoke in the troposphere. Presence of water in the smoke particles can cause a collapse of the BC agglomerates to more spherical shape (Fan et al., 2016), which should result in lowering of the depolarization ratios. This explains the low depolarization ratios for smoke from the 2nd event fires, where smoke particles were exposed to the more humid tropospheric conditions for 10 days or more before entering the stratosphere. Different aging processes in the troposphere and stratosphere could thus be the cause of differing particle depolarization ratios of smoke from the two events, although differences in fire conditions cannot be ruled out (Haarig et al., 2018; Zhang et al., 2008).

### 3.7 Decay rate of smoke

We present two estimates on the depletion rate of organics for smoke from the 1st event. Our first estimate is computed directly from the zonal mean smoke AOD (Figure 9b), suggesting a smoke half-life of $10 \pm 2$ days.

Our second estimate on the decay rate of the smoke from the 1st event is based on the CALIOP observations of individual smoke layers marked as circles in Figure 7a. We normalized the smoke signal with the local water vapor concentrations to investigate the evolution of the smoke layer composition. Water data were derived from the satellite-borne microwave limb sounder (MLS). MLS and CALIOP ran in different orbits during the smoke observations limiting the amount of collocated
data to 10 occasions for comparison, spread out in three groups over the first 50 days. We used an exponential fit and computed a corresponding half-life of $10 \pm 3$ days (Figure 9c).

The two estimates of the decay rate ($10 \pm 2$ and $10 \pm 3$ days) presented here are identical to the half-life observed for the stratospheric smoke after the 2017 North American fires ($10 \pm 3$ days; Martinsson et al. (2022)).

### 3.8 Smoke transport into the stratosphere

Stratospheric smoke from the 1st event is shown on New-Year's Eve (Figure 1), two days after the first PyroCb formations from the event. Peterson et al. (2021) coupled this transport to PyroCbs. Hirsch and Koren (2021) argued that smoke injections to the stratosphere may have occurred in the first week of January via cross-tropopause transport by convective clouds south of the fire region (38°S), where the tropopause height is lower. From the 1[st] event we do not see evidence of extensive cross-
305 tropopause transport beyond the initial PyroCb caused smoke injections in the CALIOP data (Fig 8, and supplementary). Furthermore, the temporal evolution in the UVAI (Figure 10a) indicates that most of the smoke remained north of 40°S in the days following each fire event when most of the UVAI was generated.

Magaritz-Rohnen & Raveh-Rubin (2021) suggested that cyclones and isentropic cross-tropopause transport caused smoke
transport to the stratosphere over the South Pacific Ocean in the first few days of January. This is to some extent in agreement with the UVAI (Figure 10), which increased on Jan 2-3 at 140-180°W, 35-40°S, indicating upwards transport of smoke. However, the depolarization ratios during the first week of January do not indicate cloud formation connected to the smoke layers. Furthermore, CALIOP observations show that large amounts of smoke were present in the stratosphere several days before the suggested cyclonic transport, indicating that PyroCbs were the primary cause of the direct smoke transport to the
stratosphere.

The 2nd event fires (Jan 4) positioned dense smoke layers in the mid and upper troposphere (Figure 2). Figure 11 illustrates CALIOP observations of individual smoke layer's vertical position relative to the tropopause, including layer tops, mid-points,

and bases. These three parameters all show a gradual transport of smoke from the troposphere to the stratosphere, occurring over the course of 1-2 weeks. Such transport could be caused either via self-lofting by radiation heating, via isentropic cross-tropopause transport, or via a combination of the two phenomena. We investigated potential self-lofting of smoke from the $2^{nd}$ event by studying the potential temperature of the smoke layers at their top, mid-point, and base (Figure 11). The increasing potential temperature over time indicates that they were subject to self-lofting by radiation heating, thus following the rising trend in the stratosphere, as demonstrated in Figure 7b, also in the upper troposphere. The continued addition of smoke, transported from the upper troposphere, most likely resulted in the second peak in the AOD (Figure 5). Hence, we explain the bimodality in AOD as the combined effect of rapid decay of PyroCb injected smoke, from mostly the 1st event, and slower self-lofting of tropospheric smoke from the 2nd event.

### 3.9 Long-term impact of smoke

The second AOD peak does not show as rapid decay as the first one (Figure 9a), likely due to depletion of organics during its long residence in the troposphere before the smoke from the 2nd event entered the stratosphere. The AOD evolution (Figure 9a) suggests cross-tropopause transport over the course of several weeks.

Our study indicates that smoke from the 2nd event had larger long-term impact on the stratosphere compared with the 1st event. It constituted 80-90% of the smoke signal six months after injection to the stratosphere (Figure 9a). Peterson et al. (2021) reported the opposite, namely that the 1st event injected 2-8 times more smoke than the 2nd event did. Their study focused entirely on injections by PyroCbs. Our study indicates that additional processes, acting on smoke layers deposited in the upper troposphere by pyroCbs, were more important for the long-term stratospheric aerosol load than the direct smoke injection by PyroCbs. This is in part supported by Peterson et al. (2021) who reported more blow-ups and larger area burnt for the $2^{nd}$ event. Smoke from the 2nd event was likely already depleted by photolysis before entering the stratosphere and therefore more resistant to depletion by photochemical processing. Hence, the smoke from the 2nd event fires lead to more long-term impact on the stratospheric AOD and more climate cooling.

### 4 Conclusions

The Australian wildfires in Dec 2019 - Jan 2020 caused the largest increase in stratospheric aerosol load in the southern extratropics since the large volcanic eruption of Mount Pinatubo in 1991. The long-term stratospheric AOD increase was more than three times that of the North American fires in Aug 2017, and since Pinatubo only matched by the Sarychev (2009) and Raikoke (2019) eruptions.

The AOD showed a bimodal peak in the first weeks, likely caused by the combined effect of multiple additions of smoke to the stratosphere together with photolytic loss of organics in the smoke. Smoke was added to the stratosphere from two events.

The 1st event of the fires (starting Dec 29) formed PyroCbs that injected smoke directly into the stratosphere. PyroCbs from the 2nd event (Jan 4) injected less smoke to the stratosphere but added large amounts of smoke to the upper troposphere. The stratospheric aerosol load increased rapidly, forming the first peak one and a half week after the first PyroCb injections of smoke. The AOD then dropped rapidly, likely due to aerosol depletion by photolytic loss. Upper tropospheric smoke from the 2nd event was transported to the stratosphere gradually during the course of 1-2 weeks, beginning more than a week after the

fire, causing a second peak in the AOD.

We find evidence of photochemical depletion of organics in the smoke, similar to our recent findings after the 2017 North American fires. The half-life of smoke injected directly to the stratosphere was estimated from the zonal mean AOD ($10 \pm 2$ days) as well as from compositional observations ($10 \pm 3$ days). These estimates are almost identical to our previous estimate

on the smoke half-life from the North American fires in 2017 ($10 \pm 3$ days, Martinsson et al. (2022)). This indicates that organic depletion is a commonly occurring phenomenon in wildfire smoke. Further, the rapid decay rate implies that photolytic loss has large importance for the removal of organic aerosol in the atmosphere. The rapid depletion of smoke from the 1st event leads to a small long-term impact on the stratospheric aerosol load.

Smoke from the 2nd event constituted most of the long-term impact on the stratospheric aerosol load. This was also the stronger of the two events according to the UVAI. Stratospheric smoke AOD from the 2nd event fires decreased slowly, and its morphology indicates chemical processing in the troposphere before entering the stratosphere. Particle properties (lower particle depolarization ratios) for this smoke, compared with smoke from the 1st event fires, suggest that the BC agglomerates collapsed to a more spherical state before entering the stratosphere. The particle residues remained in the stratosphere for up

to a year.

The smoke injections from the Australian fires were larger than reported in previous work and caused the largest increase in the Southern Hemispheric stratospheric aerosol load since the Pinatubo eruption. We argue that wildfire smoke has become an important part of the stratospheric aerosol with climate impact comparable to moderate sized volcanic eruptions. Wildfires

are in part natural and in part caused by humans. Future fires are projected to become more intense and frequent due to climate change. Hence, the climate impact of stratospheric wildfire smoke must not be neglected in future climate projections.

**Data availability**

CALIOP V4.10 and 4.11 lidar data are open access products (https://search.earthdata.nasa.gov/search?fp=CALIPSO, Hostetler et al., 2006). OMPS-LP aerosol extinction coefficients (Taha et al., 2021) were accessed via

https://disc.gsfc.nasa.gov/datasets/OMPS_NPP_LP_L2_AER_DAILY_2/summary. OMPS-NM V2.1 UV aerosol index

(Torres, 2019) was obtained from https://worldview.earthdata.nasa.gov and https://ozoneaq.gsfc.nasa.gov/data/omps/#prods=84,101. H2O data from MLS were obtained from https://disc.gsfc.nasa.gov/datasets?page=1&keywords=ML2H2O_005 (Lambert et al., 2020).

**Author contribution**

JF designed the study, performed most of the data analysis, and wrote most of the paper. BGM undertook data analysis on individual smoke layers and wrote part of the methods section. MKS produced the supplementary. All authors contributed to discussions and commented on the manuscript.

**Competing interests**

The authors declare that they have no conflict of interest.

**Financial support**

The Swedish research council for sustainable development, Formas, contract 2018-00973 and 2020-00997
Swedish National Space Agency, contract 2022-00157

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

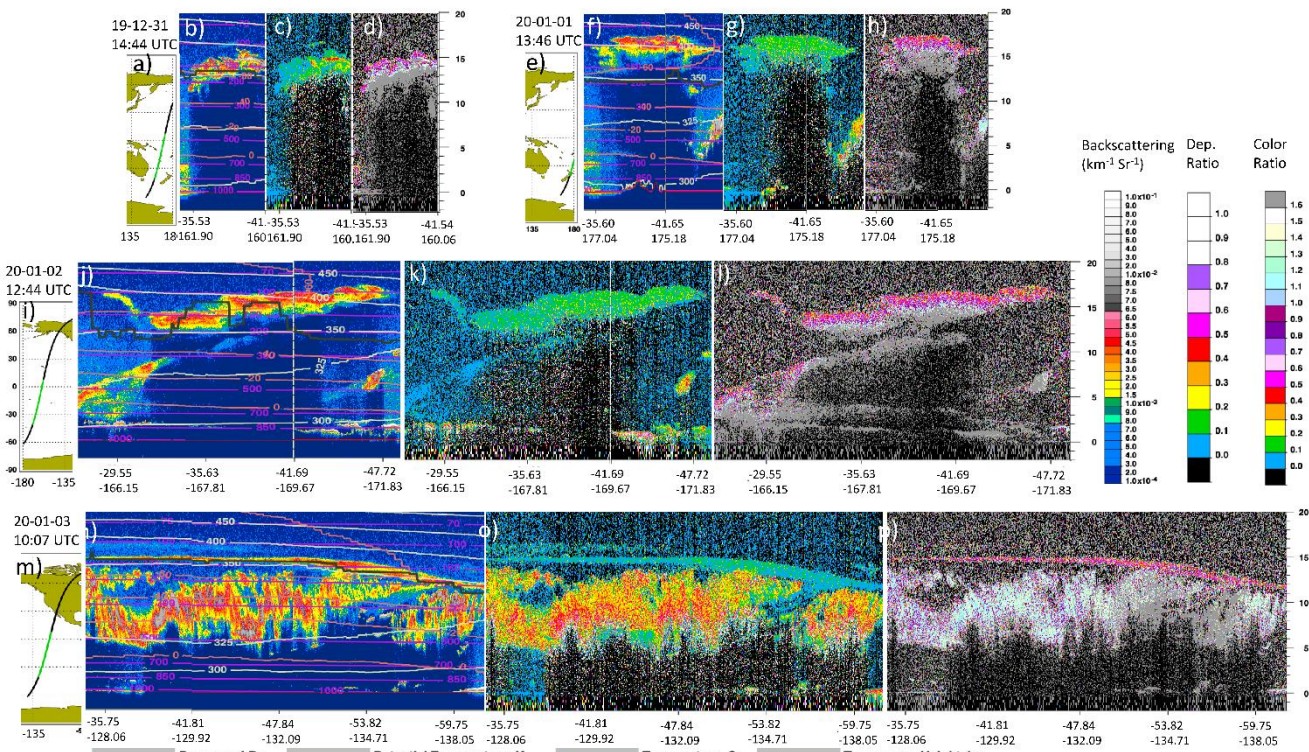

Figure 1. CALIOP curtains from early overpasses of the region affected by the 1st fire event that started on 2019-12-29. Left panels show attenuated backscattering, mid panels, the volume depolarization ratios, and right panels the attenuated color ratios. Strong attenuation of the lidar beam is indicated by the dark blue colors below smoke layer, as well as through the higher color ratios at the bottom of the smoke layers. Meteorological parameters are marked with lines; pressure levels (purple), potential temperatures (white), temperatures (orange), and tropopause heights (grey). See the supplementary for further curtain plots.

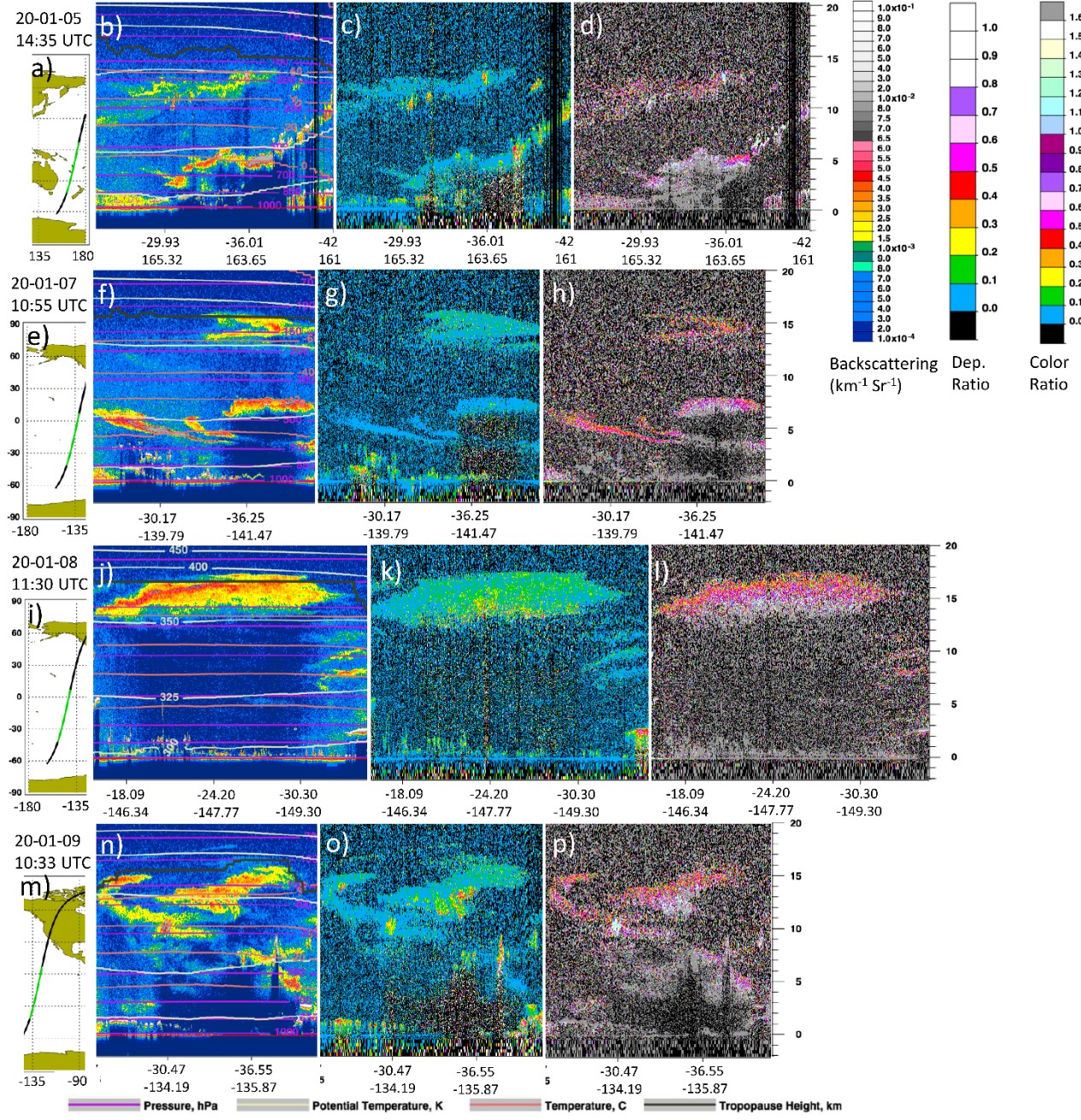

Figure 2. Same as Figure 1, but for smoke layers from the 2nd fire event (2020-01-04).

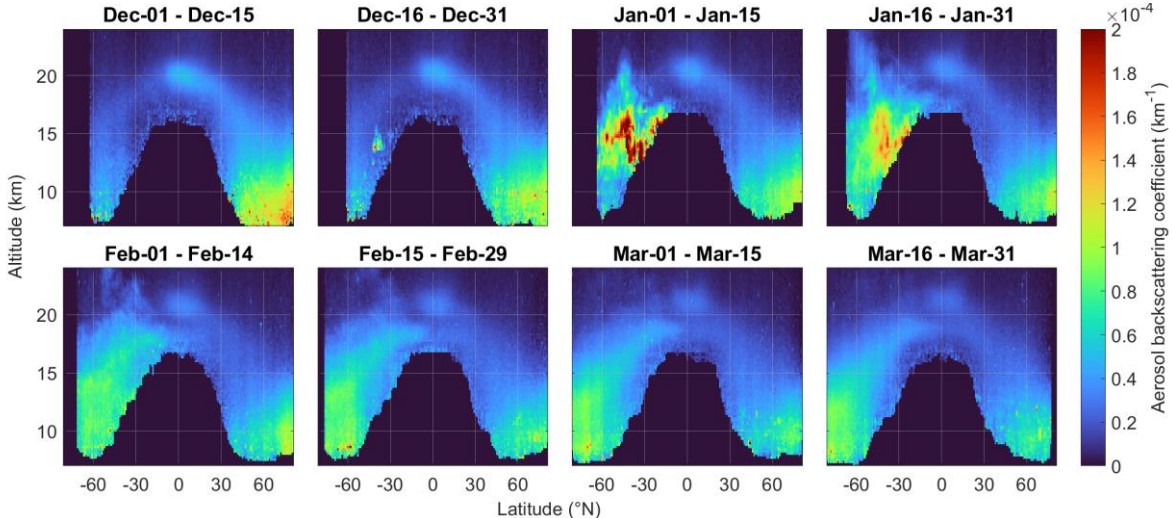


**Figure 3 | Latitudinal and vertical distribution of smoke in Dec 2019 – Mar 2020.** CALIOP zonal mean aerosol backscattering coefficients during the first three months after, and one month preceding, the first stratospheric injection. This parameter can be viewed as an optical version of aerosol concentration.

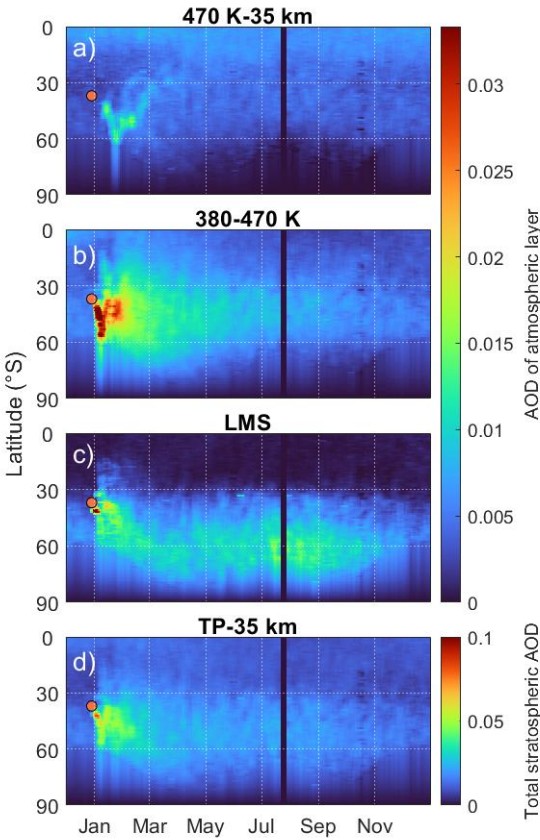


**Figure 4 | Latitude and time distribution of the stratospheric aerosol load. Aerosol optical depth (AOD) in three stratospheric layers and in all layers combined. The upper (a) and mid (b) layers are the deep and shallow branches of Brewer-Dobson circulation, and the lowest layer (c) is the lowermost stratosphere. The orange dot shows the approximate latitude and the time of the 1st fire.**

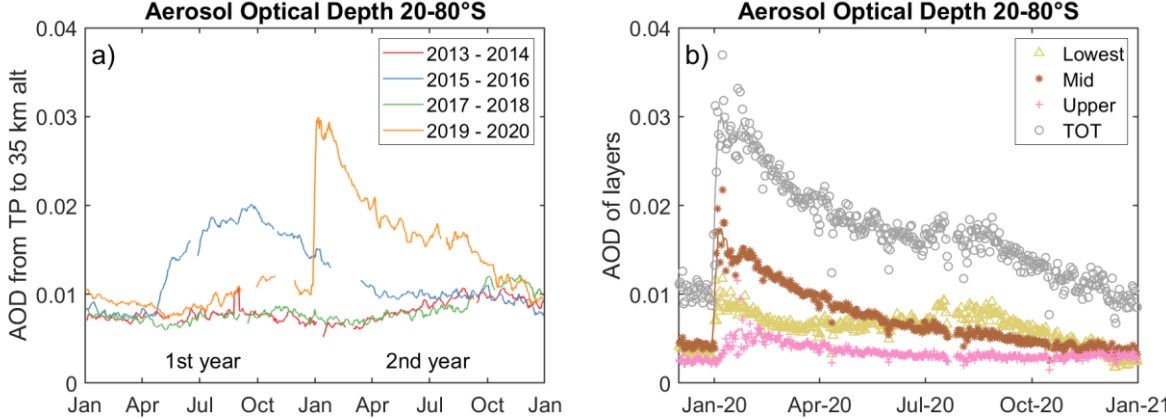


**Figure 5 | Wildfire impact on the stratospheric aerosol load in the southern extratropics. a) The aerosol optical depth (AOD) from CALIOP in 2013-2020 divided into two-year-steps, illustrating the impact of the Australian wildfire in 2019-2020, together with that from the Calbuco eruption in 2015-2016, and background levels 2013-2014 and 2017-2018. b) the stratospheric AOD from CALIOP during Dec 2019 – Jul 2020 separated into the three layers used in Figure 4. Lines mark 8-day smoothed AOD data, and symbols**

**are daily means of the LMS (yellow), shallow (brown) and upper (pink) BD-branches, and the total stratospheric AOD (grey).**

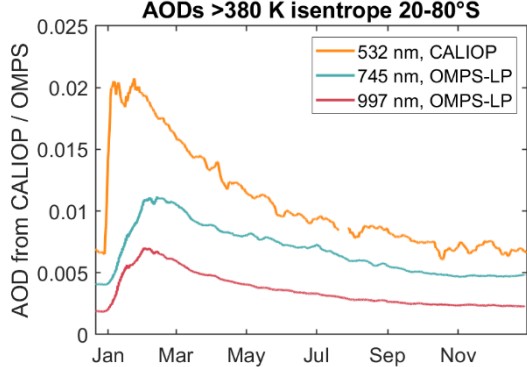

**Figure 6 | Wildfire impact on stratospheric AODs at three wavelengths in the southern extratropics from CALIOP and OMPS-LP. The stratospheric column above the 380 K isentrope column at 20-80°S.**


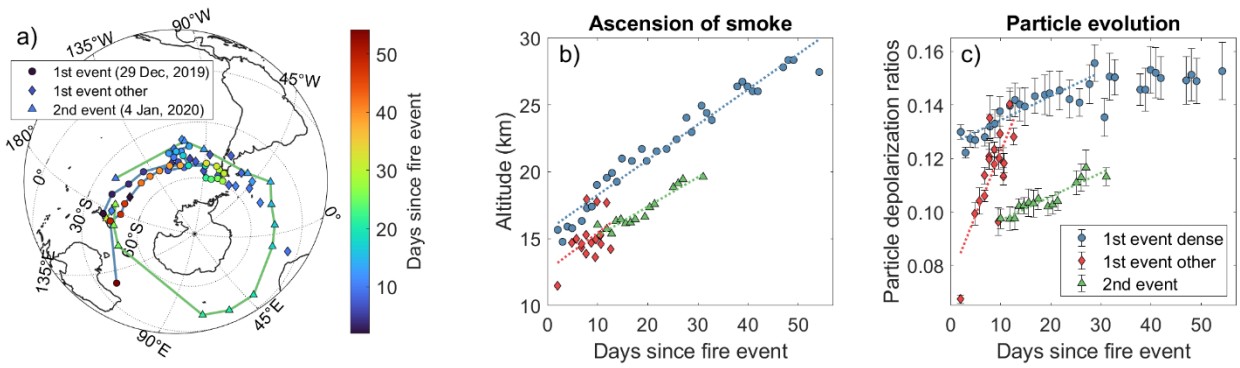

**Figure 7 | The smoke transport and smoke particle evolution in the stratosphere. a) Geographical transport of the smoke from the two injection events. b) Ascension of the smoke clouds, and c) temporal evolution of the particle depolarization ratios for the two injection events. All data taken on individual smoke clouds in the stratosphere. Circles mark data taken in the dense isolated cloud from the 1st event fires, diamonds are other data from the 1st fire, and triangles mark smoke data from the 2nd event fires.**


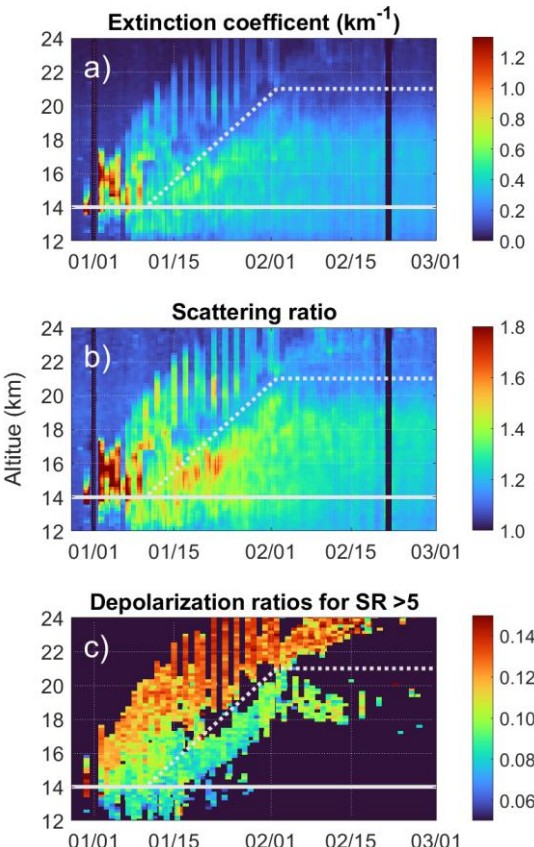

**Figure 8 | Separation of the smoke from the two events of the fires. Stratospheric zonal mean (20-80°S) a) Extinction coefficients, b), scattering ratios (SR), and c) depolarization ratios where SR values are higher than 5. The dashed and full lines represents the separation line of smoke for the two events of the fire, and the minimum altitude used to compute the AODs for the two events of fires.**

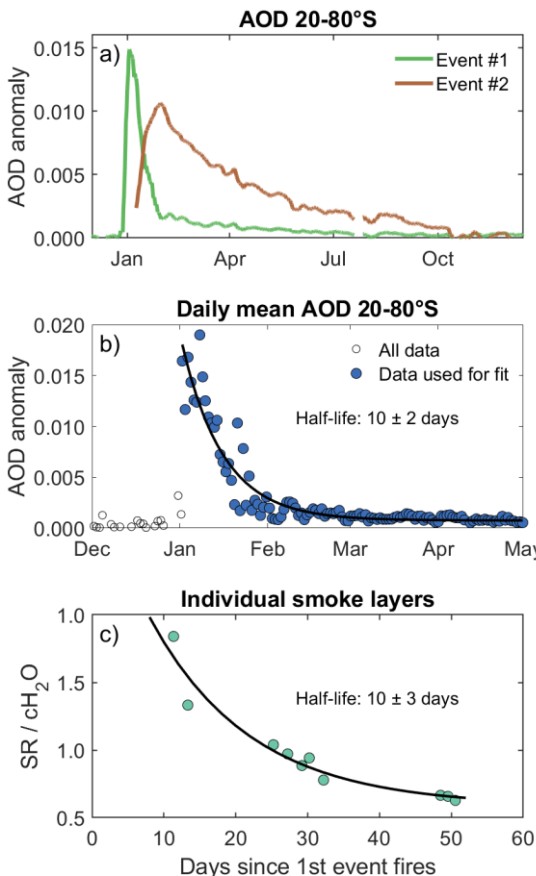

**Figure 9 | Smoke decay in the stratosphere. a) 8-day running mean of the background subtracted stratospheric zonal mean AOD at 20-80°S above 14 km altitude for the 1st and 2nd events, respectively. b) Daily means of background subtracted AODs for the 1st event only, and c) smoke data from individual smoke layers from the dense isolated smoke from the 1st event (scattering ratios, SR, from CALIOP) normalized with water vapor concentrations (cH2O, from MLS). The exponential fits correspond to a smoke half-life of 10 ± 2 (b) and 10 ± 3 days (c).**

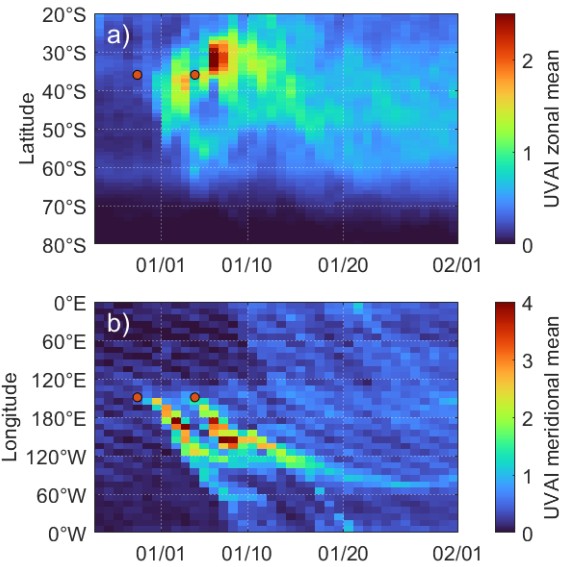


**Figure 10. | Evolution of the UV aerosol index (UVAI) during the first weeks after the fire events for a) zonal means and b) meridional means. The locations and dates of the events are marked by orange dots. The lower UVAI range in a) stem from the method of area weighting data.**


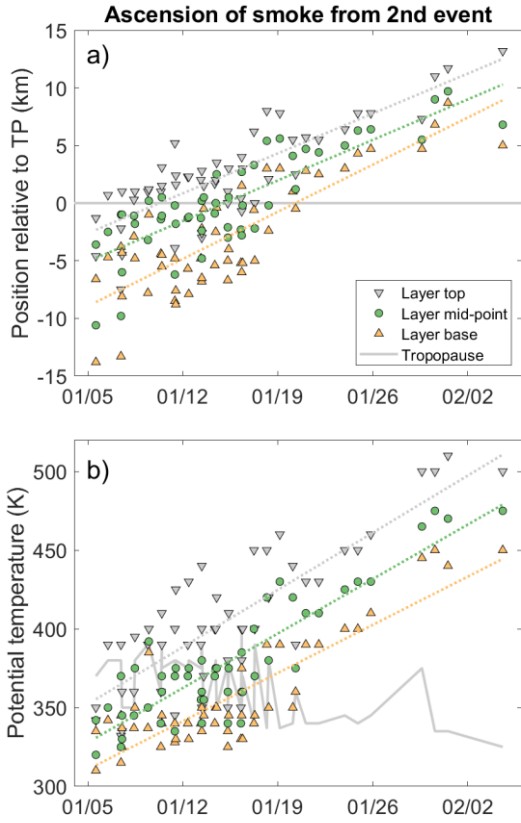

**Figure 11. | Ascension of 2nd event smoke layers for a) smoke layer position relative to the tropopause (TP), and b) smoke layer potential temperature. Circles mark the layer midpoints, downward and upward pointing triangles mark the layer tops and bases, and the grey solid line marks the tropopause. Data were retrieved from CALIOP nighttime curtain plots (see supplementary for further details), where layers with layer tops below 8km where included in the graph. Dashed lines show regression lines for layer tops (grey), midpoints (green), and bases (yellow).**

**Table 1 | Wildfire and volcanic impact on aerosol optical depths and radiative forcing. The one-year AOD increase in the extratropics (20-80°N/S) after the largest volcanic eruptions and wildfires since 2006 compared to the Australian wildfires.**

| Date | Location | | Event name | 1 y AOD incr. | 1 y AOD incr. CALIOP |
|------|------|------|------------|----------------|------------------------|
| | Lon. | Lat. | | CALIOP | rel. Aus. Fires |
| 2008-08-07 | 176°W | 52°N | Kasatochi | 0.0059 | 63% |
| 2009-06-12 | 153°E | 48°N | Sarychev | 0.0090 | 97% |
| 2011-06-12 | 42°E | 13°N | Nabro | 0.0057 | 61% |
| 2015-04-23 | 73°W | 41°S | Calbuco | 0.0080 | 86% |
| 2017-08-12 | 120-125°W | 49°N | N.Am.Fires | 0.0027 | 29% |
| 2019-06-22 | 153°E | 48°N | Raikoke | 0.0104 | 110% |

| | | | | | | |
|---|---|---|---|---|---|---|
| 2019-12-29 - 2020-01-04 | | 147-151°E | 34-38°S | Aus. Fires | 0.0093 | - |
