# Peer review of "Short and long-term stratospheric impact of smoke from the 2019/2020 Australian wildfires"

_EGUsphere, 2023_

## Referee Comment (RC1)

**Review of Friberg et al., 2023**

The manuscript of Friberg et al. focuses on stratospheric impact of smoke from the 2019/2020 Australian wildfires. They define two events of smoke injection for their study and find a half-life time of the smoke of 10 days. They claim it to photochemical processing of organic aerosol. The manuscript should contain more references and a more convincing argumentation that photochemical processing played a significant role in the decrease of smoke AOD – in the current version of the manuscript it is not convincing. Below, the main concerns are listed in more detail:

Main concerns:

What lidar ratios at 532nm are used to evaluate CALIOP observations for volcanic sulfate and Australian smoke?

Please check the papers Ohneiser et al., ACP, 2020 and Ohneiser et al., ACP, 2022. These papers may serve as reference for all the satellite observations (CALIOP, OMPS, AOD).

You argue that the life-halftime is 10 days because the smoke particles dissolve. How could Ohneiser et al., ACP, 2022 observe the smoke then for 1-2 years after the emission? Please comment on that.

20°-80°S mean, this is from the tropics to the polar region. What about 30°-70°S? That can be better compared with the reference lidar observations above Punta Arenas at 53°S.

Referencing seems to be arbitrary, especially in the Introduction. A good overview: What is already available (regarding this record-breaking event) together with appropriate references would be helpful! Which gaps are left and filled by this paper?

Specific concerns:

9 lofted

12 It was so inhomogeneous, how could one determine half-life? Decay is also a function of horizontal meridional dispersion.

21 Source

22 define extratropics

34 References

41 eruptions without `

51-56 Aging does not just remove/decrease smoke mass. How is that linked to the findings in Ohneiser et al., 2022: 20 months measurements, slow decay or even Canada, Siberia 5-8 months residence time. What about dispersion? Please add references.

73 kilometers

80 brackets away around Martinson source

80 define SH

104 show instead of shows

170-171 AOD CALIOP: What lidar ratios are used? Background and Calbuco and Australian fires. Please compare all these also with ACP, 2020 and Ohneiser et al., ACP, 2022.

216 Figure 2 instead of Fig 2

226 Decreasing depolarization ratio: put it into context with the findings in Baars et al. 2019 (for example) who found decreasing depolarization ratios after stratospheric smoke injection

233-244 Same as earlier comments: What about sedimentation, what about dispersion, why can the smoke be observed for such a long time, when you write that half-life is 10 days?

239-244 typically, the findings tend to a decreased BC fraction with time as the coating with organics increases with time. This is in contradiction with your findings please comment on that.

273 you could add Ohneiser et al. 2023 in that context (Ohneiser, K., Ansmann, A., Witthuhn, J., Deneke, H., Chudnovsky, A., Walter, G., and Senf, F.: Self-lofting of wildfire smoke in the troposphere and stratosphere: simulations and space lidar observations, Atmos. Chem. Phys., 23, 2901–2925, https://doi.org/10.5194/acp-23-2901-2023, 2023.)

282-283 Rieger et al., GRL, 2021 show the opposite. Most of the smoke went south to 70-80°S. Why should the efficient transport pathway be to the north? In your Fig. 4, also most of the smoke is located south of 60°S.

312 Aging works also the other way around: Condensation of gases onto the particles, therefore they can have a long lifetime

365-373 These three literature entries are not included in alphabetical order

Fig. 1: Too busy, no text in the figure readable, too many subfigures, no legend, no continent boundaries visible, no latitude and longitude range visible

Fig. 2: X axis text missing, too many figures, too small text, use (a), (b), (c)… in the figure to be able to refer to different parts of the figure. The figure organization is very confusing. Always show the same right below each other and not all the subfigures in different sizes. Please use less subfigures if not every figure is important for the paper.

Fig. 3: Same as Fig. 2, too small, too confusing, too much.

Fig. 5: What is the orange point? What about self-lofting impact?

Fig. 6: What was the used Calbuco lidar ratio and Australian fire lidar ratio? 30-70°S would be better.

Fig. 7: How are these results in agreement with the only reference dataset in Ohneiser et al., ACP, 2020 and Ohneiser et al., ACP, 2022? There could be saturation effects.

Fig. 10: What about using latitude subregions: 30-40°S, 40-50°S, 50-60°S, 60-70°S?

---

## Referee Comment (RC2)

Review of Friberg et al., "Short and long-term stratospheric impact of smoke from the 2019/2020 Australian wildfires." (Hereafter "F23").

Reviewer: Mike Fromm

**Overview**

F23 present a new satellite-observation-based analysis of the Australia New Year (ANY) 2019/20 fire/pyroCb event. Their aim is to quantify the stratospheric smoke burden and evolution, resolving the two separate contributions from the December and January phases of ANY. There are three distinctive and new contributions herein. 1. determination of smoke AOD decay via photolysis, 2. non-pyroCb-pathway stratospheric pollution by the January-phase plume, yet 3. dominant influence of the January-phase smoke on the overall ANY stratospheric perturbation. In this pursuit, F23 employ aerosol data from nadir-viewing OMPS absorbing aerosol index, limb-view OMPS-LP aerosol extinction, CALIOP backscatter, and $H_2O$ data from MLS.

This is a pursuit worthy of study and natural for ACP. The ANY case, individually and by comparison to other stratospheric smoke and volcanic events, is still imperfectly understood and the subject of varying accounts regarding transport pathway, physical evolution, composition, and radiative impact. This study intends to reduce the uncertainties in at least two important ways.

However, F23 have taken on a very complicated scenario, involving continuous smoke generation in southeast Australia in late 2019 and early 2020, punctuated by episodes of pyrocumulus (pyroCu) and pyrocumulonimbus (pyroCb) activity, in a dynamical meteorological setting. I was not convinced by F23's core new finding—that the January-phase smoke was injected only into the troposphere and didn't rise into the stratosphere until ten days post event. F23 build their case with a subset of their data items that under-samples the downstream smoke to the detriment of accurate plume-height and transport-pathway characterization. F23's analysis of aerosol-layer stratification in their illustrations is vague. In addition, their considerable reliance on OMPS UVAI maps (Figure 1) to aid in plume distinction as the smoke blends together is problematic.

F23 comes on the heels of two other papers attributing ANY's unprecedented mass of stratospheric smoke to a non-pyroCb pathway. Hirsch & Koren (2021) found no pyroCbs, only random oceanic thunderstorms to be the pathway. Magaritz-Ronen, & Raveh-Rubin (2021) concluded that pyroCbs did not suffice, and that a single Pacific-Ocean synoptic-scale cyclone on ~2 January performed the task. Now F23 attribute the preponderance of ANY stratospheric smoke to tropospheric self-lofting as the pathway while briefly dismissing the above two explanations. Given that Peterson et al. (2021) demonstrated clusters of pyroCbs in both December and January that penetrated the tropopause, and unprecedented UVAI plumes immediately following each impulse, is it F23's position that neither the pyroCb nor the cyclone nor thunderstorm pathway can explain the long-lasting stratospheric smoke observations? If so,

their analysis of satellite-data, transport, and lofting needs to be much more exacting than what is presented herein.

My assessment is that if F23 can mount a convincing argument of a diabatic troposphere-to-stratosphere pathway for the January-phase plume, the balance of material could become defensible and thus merit publication. But as will be elucidated in my report, F23's analysis of the December and January ANY smoke impulses is unclearly developed and plausibly incorrect. Hence, I recommend major revisions of this manuscript after considering all the concerns I list below.

**General Concerns**

A major concern of mine is that F23 discounted a crucial element of the pyroCb "smokestack" phenomenon. The ANY event, like all pyroCb cases, involves the direct injection of smoke, ice, and biomass-burning gases to the top of the pyroconvective column. Like all convective exhaust, these materials disburse at column-top altitudes. These injection heights are quantified in the case of ANY and several other pyroCb events (such as Black Saturday, PNE, Chisholm, etc.) by weather-radar reflectivity and/or infrared brightness temperature image data. In the case of each, it is known that the column-top estimates are generally conservative, i.e. low biased. This has been documented (Fromm et al., 2021, **https://doi.org/10.1029/2021JD034928** and references therein). Hence the source term for any such pyroCb event is reliably known to the extent that the pyroCb clouds are thus characterized. In the case of ANY, Peterson et al. (2021) provided an exhaustive accounting of all of the December- and January-phase source terms. Both the December and January phases embodied plumes that topped out in the LMS. Detection of those plumes post-pyroCb are then subject to spotty sampling by any and all satellite instruments. It is regularly the case that pyroCb-plume case studies involve these imperfect satellite data items convolved with various time lags between injection and ideal sampling. Because these fire events also involve somewhat continuous emissions throughout a wide range of vertical transport that the downstream plume picture is embodied by thick and thin smoke plumes from the surface to the topmost injection heights. Incomplete/imperfect sampling of these plumes can be suggestive of a multitude of pathways and processes. In the case of ANY, we can state with certainty that the December and January pyroCbs created smoke plumes that topped out above the tropopause ambient on the active dates. In the case of the 4 January pyroCbs, echotops exceeded 16 km (and Θ exceeding 380 K). Consistent plume heights were observable and traceable to these pyroCbs, 2-3 days post pyroCb. But even if such downstream sampling was non-existent or woefully incomplete, it wouldn't change the underlying veracity of the "smokestack" pathway and endpoint. Therefore, the challenge for identifying any other contributing pathway to the stratosphere in the case where direct injection is clearly established is thereby heightened. F23's Figure 2 and 3 appear to show confirmation that both December and January ANY smoke matches the smokestack expectation. Additional support for this end-to-end connection is provided in the supplement at the bottom of this review. F23 are encouraged to weigh this argument and evidence, and consider how it informs their approach.

Figures 1 (daily UVAI maps for ~5 weeks), 2 and 3 (ensembles of CALIOP curtains) are used to make the case that the December-phase of smoke injections immediately reached the lowermost stratosphere (LMS) but the January-phase smoke did not. As one would expect, the smoke emissions over the weeklong pyroCb event generated plumes that were straightforward to separate for a brief period, then became indistinguishable based on the UVAI alone. F23 blended the UVAI data with selected nighttime CALIOP curtains to attribute a portion to December and the balance to January. They discuss how they evaluated plume transport to connect these elements with the two phases, but that transport was only inferred via the day-to-day change in the UVAI plume positions. I could not find any additional tool for source-receptor connection such as Lagrangian trajectories. By its nature, the UVAI depends largely on AOD and altitude such that a low, dense layer's UVAI could be indistinguishable from a high, optically thin layer. It is obvious from F23's Figure 2, 3 that multiple layers, from the free troposphere to the LS, were the rule downstream of ANY. The flat color gradation used in the UVAI maps limits any meaningful discernment of low/high, dense/thin features. And as mentioned above, after the first week of January, the previously distinct UVAI plumes become inseparable. F23 recognize this and qualify their analysis with their selection of CALIOP curtains matched to the UVAI features. However, every CALIOP curtain used by F23 is from a nighttime orbit segment. These are systematically ½ day offset from the UVAI-measurement time. This might be OK if there is only one layer at stake and that layer is governed by very light winds. But given the true complexity of the smoke layering and natural wind-shear involvement, the association of the CALIOP features with the UVAI is wholly uncertain. F23 might have invoked other imagery-based retrievals available at night, such as IR-based CO, but they did not. F23 might have invoked daytime CALIOP curtains, but they did not. Hence the reader is inadequately informed about the true association of the CALIOP and UVAI features.  On its merits, this complicated ANY smoke event requires a more rigorous and precise accounting between the CALIOP/UVAI and the two ANY injection phases.

For defensible reasons, F23 do not employ OMPS-LP aerosol extinction data as they do CALIOP for the assessment of the nascent plume altitude. However, this may have been a missed opportunity, given the limb-view data's coincidence with the OMPS-NM UVAI.  While acknowledging F23's cited concerns about OMPS-LP utility in the presence of optically thick plumes, its natural combination with the UVAI can and does allow for a confident characterization of LMS plume-top. Moreover, it will be shown that such a combination reveals a finding of the LMS position of the young January-phase plume days before F23's conclusion.

F23's main analysis leading them to conclude that the January-phase pyroCb smoke was almost totally relegated to the troposphere (until it diabatically lofted across the tropopause ~13 January) exploits the CALIOP curtains shown in Figure 3 and selected point observations in Figure 8. Figure 3 has 12 panels, curtains from 5-10 Jan. Most panels show dense smoke straddling or above the tropopause. Presumably F23 do not attribute these to 4 January. But it is unclear how they interpret Figure 3. The text discussion calls out Figure 3 but does not offer any in-depth explanation of the various aerosol features. On its face then, Figure 3 seems to

contradict F23's premise. A sufficiently detailed discussion, commensurate with the many Figure 3 panels and the multitude of aerosol layers therein, is essential. By the way, this recommendation also applies to Figure 2 and attendant analysis.

F23 exploit CALIOP depolarization ratio in an interpretation of particle morphology. Two separate threads are presented. 1. They evaluate a temporal increase of depolarization in the stratospheric smoke, in combination with AOD decay, as evidence of photolysis-imposed loss of organic mass fraction. 2. Small depolarization of the January-phase tropospheric smoke as a sign of aging in a humid environment with a transformation to more spherical particles.

Regarding point 1, two questions arise. First, if the particles are losing mass, one might expect to see a change in CALIOP color ratio commensurate with particle-size reduction. F23 introduce CALIOP attenuated color ratio in Figures 2 and 3, so it is natural to ask if they analyzed the temporal variation of that quantity in relation to depolarization ratio. If so, those results would be to F23's advantage to report. If not, it is reasonable to suggest that F23 perform this analysis to more fully support their line of argument. Also, Baars et al. (2019; https://doi.org/10.5194/acp-19-15183-2019) showed that the 2017 Pacific Northwest stratospheric pyroCb plume decreased in depolarization over the course of ~3 months. Might F23 cite that work and comment on the implications of such transformation? Would that be consistent with a resupply of organics or other shell material on the BC core? Is there evidence of CALIOP depolarization ratio decline after the flattening shown in Figure 8?

Regarding point 2, is it reasonable to speculate on how particles that have a collapsed, nearly spherical BC core would become less spherical during the aging in the stratosphere? What is the evolution of the depolarization ratio of the tropospheric smoke?  Is there a perceptible decline in time, consistent with the proposed aging?  A more concrete concern regarding this point is that F23 generalize the ambient condition of the tropospheric January-phase smoke as "humid." While it is indisputable that the stratosphere is dryer than the troposphere, the speculation here about the ambient conditions implies (to me) that the smoke particles are far below the relatively dry UT. Is it F23's expectation that the January plume was largely lower than the UT (as several of the lower, dense layers in Figure 3 are) and thus lofted by several km before entering the stratosphere? See my earlier comment about the need for a much more precise treatment of the features in Figure 3.

I was confused by the general treatment of the December and January plume discussion. It appears that F23 are dealing separately with the vortical, rising plume elements and the overall ANY stratospheric smoke.  Given that the two compact plume sub-elements represent a minor contribution to the overall AOD of the ANY smoke, the discussion of the global AOD evolution of the two phases seems to be unrelated to the compact-feature evolution. Perhaps I did not catch on to the interplay between the two themes. Please point out to me what I am missing, or otherwise add some discussion material to help the reader appreciate the two seemingly independent themes.

F23 claim that the January-phase smoke plume ascended gradually from the troposphere to the stratosphere over the period of 4-14 January. Figure 3, which chronicles the perceived 4 January smoke via CALIOP imagery terminates on 10 January. There are no additional granular CALIOP curtains filling the gap to 14 January. Presumably F23's suggested rise of the smoke to the tropopause would be discernable beyond 10 January (Figure 3). The reader should have the access to the full interpretation of the smoke evolution to be convinced of F23's evidence. The 4-day gap in this timeline stands as a barrier to this understanding. It is essential for F23 to present this entire timeline or explain how the evidence they show in Figure 3 is sufficient to secure their claim.

Section 3.8, "Smoke transport to the stratosphere" is perhaps the most consequential section of this paper. Yet it is populated by weak, confusing, and misleading points. Next, these are shown in bold followed by my reaction in plain text. **"The NA wildfires in 2017 showed that self-lofting by radiative heating of the dense smoke layers caused smoke to rise from the UT into the LMS (e.g. Khaykin et al., 2018; Peterson et al., 2018)."** This is an apparent mischaracterization of these two cited papers. Neither made that argument, or even hinted at it. The diabatic lofting observations they showed all started at stratospheric altitudes. If I missed these points in the two cited papers, please point them out to the reader. **"Ohneiser et al. (2021) suggested self-lofting of smoke from the midtroposphere as cause of extensive smoke layers in the Arctic stratosphere during in the end of 2019 and beginning of 2020."** F23 might consider balancing this statement by citing Boone et al. 2022; https://doi.org/10.1029/2022JD036600 ), who categorically refuted Ohneiser's characterization of extensive stratospheric smoke at that time. **"Hirsch and Koren (2021) argued that smoke injections to the stratosphere may have occurred in the first week of January via cross-tropopause transport by convective clouds south of the fire region (38°S), where the tropopause height is lower. However, the temporal evolution in the UVAI (Fig. 11a) indicates that most of the smoke remained north of 40°S, and only a minor portion was located south of 45°S."** But Figures 1 and 2 show that there was abundant smoke south of 45S in late December and the first week of January. This comment is in no way an endorsement of Hirsch & Koren's claim; it is meant to point out that F23 show data that are in apparent direct conflict with this statement. **The 2nd event fires (Jan 4) positioned dense smoke layers in the mid and upper troposphere (Fig. 3).** Figure 3 shows tropopause-level and/or LMS smoke in 9 of the 12 panels. As I commented above, F23 do not examine Figure 3 (or 2) in granular detail or perform an explicit feature-by-feature source attribution. The reader is given no framework on which to match the above statement with the illustrations. **We see evidence of vertical transport during the following week.** What is this evidence? If this is referring to Figure 3, the evidence is not at all clear to me. For the reader to see F23's evidence, a much more detailed analysis of Figure 3 is required. **The generally low depolarization ratios (<0.10) are not indicative of cloud formation.** What is the relevance of this statement? Since this is referring to Figure 3, it is apparently meant to compliment F23's argument that this is smoke from 4 January Australia. Moreover, there are several large non-cloud features in Figure 3 that have depolarization ratio >>10%. Hence, this statement requires expansion toward a full characterization of the various scenes provided. **"…we suggest self-lofting by radiation heating and isentropic cross-tropopause transport as the cause of transport to the stratosphere for smoke from the 2nd**

**event, thus following the rising trend in the stratosphere, as demonstrated in Fig. 8b, also in the upper troposphere."** "We suggest" implies a considerable amount of uncertainty (as opposed to, e.g. "we find" or "we demonstrated"). Prior to this statement F23 claim to see "evidence" of self-lofting in the troposphere. So, the reader needs to know if their evidence is inconclusive, which would leave F23 in the position of just "suggesting" it. If their evidence is shown to be conclusive, then their presumed position would be that tropospheric self-lofting was demonstrated. Please explain the claimed evidence and discuss their confidence level in the interpretation. This statement also mentions isentropic cross-tropopause transport as a pathway to the stratosphere. They made no prior claim of finding evidence of this for the January-phase smoke. Their only treatment of this pathway was in reference to and partial dismissal of Magaritz-Rohnen & Raveh-Rubin (2021). Hence, it is unclear what justification there is for the suggestiveness of isentropic cross-tropopause transport. Please elaborate, or consider dispensing with this suggestion.

**Targeted Concerns**

**L198-200: "The 2nd fire event occurred on January the 4th, but smoke from this event showed only little immediate stratospheric influence (Fig. 3) in line with observations by Peterson et al. (2021)."** What are the observations from Peterson et al. that support this statement?

**Section 2.3:** In this introduction of the MLS water vapor, it is called for F23 to cite Schwartz et al. 2020; 10.1029/2020GL090831) who did an in-depth analysis of the ANY MLS $H_2O$ data.

**Figure 1:** How is the AI color scaled? What are the ascending orbits? Why are they shown? All the panels have grainy resolution. It is very difficult to discern all the additional layer features.

**Figure 1.** I was not able to find a thorough description of how the green-line demarcation between December and January phases is justified. Please explain in a way that applies to every figure panel.

**Figures 2 and 3:** The 532 nm attenuated backscatter panels all have the meteorology-data overlay. These are not described in the captions or main text. Moreover, many of the panels are cropped such that the isoline labels are not displayed. If the meteorology data are necessary, they should be appropriately labeled and exploited in the main text. If they are non-essential, please consider reducing clutter by showing just the lidar data.
Define "main layer" and "minor layer".

**Figure 8b:** The earliest triangle is ~10 days after the fire event. But the earliest triangle on the map is not where the phase-2 plume was on 14 Jan (10 days post event). This point appears to

be close, geographically, to the position on or about 7 Jan when Khaykin et al. started following this plume element. What is the date of this earliest triangle? If it is indeed ~7 January, then the triangles in Figure 8b are offset incorrectly. If instead the x-axis is relative to the December phase alone, this should be explained.

**L287-288. "However, the depolarization ratios during the first week of January does not indicate any frequent cloud formation connected to the smoke layers."** Meteorologically, what is meant by "frequent cloud formation?" What are the smoke layers to which F23 are paying attention? The reader has no basis on which to accept this assertion. Please bolster it with a callout to and detailed analysis of Figure 3, or providing an additional figure in support.

**Conclusions, L322. "Smoke was injected to the stratosphere from two events."** The first event was attributed by F23 to the December pyroCbs. What was the second stratospheric injection event? If F23 maintain the claim that the January impulse was strictly tropospheric, should this sentence be amended? If the second stratospheric injection event was from the January pyroCbs, this contradicts the arguments that are the basis of F23's findings. Please clarify.

**Technical Points**

**Figure 2. "Four days CALIOP curtains…":** Insert "of" before "CALIOP."

**Figure 2 caption. "The day of the fire…":** What is meant by this? Should it be "day of the pyroCb? Considering that this was a multi-day pyroCb phase, it would be advisable to revise this terminology.

**Figure 2 caption. "malfunction":** A description of the malfunction is called for, along with a citation so the reader knows what the anomaly is.

**Figure 2, and throughout the document. "29 Dec.":** The first phase of the ANY pyroCb event was a multi-day affair, 29-31 December. Please consider describing this phase in that way.

**Figure 3 caption. "Same as figure…":** Missing Figure number.

**Figure 9a:** A scaling factor for the extinction coefficient is missing.

**Figure 9c: "1.0" on the color bar:** Should be "0.10"

**3.2 Compared to volcanism:** Please state what is being compared to volcanism in this section heading.

**Supplement**

[Figure]

**On 7 Jan, OMPS-LP sampled the core of the coincident OMPS-NM UVAI plume.**

VIIRS/OMPS ORbit

[Figure]

**Quicklook extinction curtains for this orbit.**

[Figure]

[Figure]

[Figure]

**Trajectories lead back to SE Australia on 4 Jan**

[Figure]

NOAA HYSPLIT MODEL
Backward trajectories ending at 2300 UTC 07 Jan 20
GDAS Meteorological Data

Job ID: 121653        Job Start: Mon Mar  6 14:39:23 UTC 2023
Source 1 lat.: -28.300000  lon.: -147.000000  hgts: 15500, 16500, 17500 m AMSL

Trajectory Direction: Backward     Duration: 88 hrs
Vertical Motion Calculation Method:     Model Vertical Velocity
Meteorology: 0000Z 1 Jan 2020 - GDAS1

**GDAS T profile at OMPS location/time.**

**OMPS-LP "Cloud Top" at 17.5 km, above cold-point tropopause**

[Figure]

**OMPS-LP "Cloud Top" at 4oo K.**

**Clearly in LMS.**

[Figure]

[Figure]

24 hours later.

CALIOP and OMPS-LP coincidence.

Both sample the AI core.

[Figure]

Aerosol Extinction Values Retrieved at 997 nm($km^{-1}$)

[Figure]

[Figure]

[Figure]

**24-hr back trajectories from CALIOP lead back to prior day's AI max.**

---

## Author Comment (AC1)

We thank the reviewers Kevin Ohneiser and Michael Fromm for all constructive comments and that helped us improving our manuscript. The reviewer comments are reported in black text, after which you find responses in this blue color. Changes in the manuscript are shown by 'Trach Changes' by red underscored text.

The manuscript of Friberg et al. focuses on stratospheric impact of smoke from the 2019/2020 Australian wildfires. They define two events of smoke injection for their study and find a half-life time of the smoke of 10 days. They claim it to photochemical processing of organic aerosol. The manuscript should contain more references and a more convincing argumentation that photochemical processing played a significant role in the decrease of smoke AOD – in the current version of the manuscript it is not convincing. Below, the main concerns are listed in more detail:

**Main concerns:**

What lidar ratios at 532nm are used to evaluate CALIOP observations for volcanic sulfate and Australian smoke? We computed the effective lidar ratios of individual smoke layers using the methods described in Martinsson et al. (2022), where we also corrected the CALIOP data for attenuations by molecules (including ozone) and particles. The mean values used for smoke from the Dec 29 fires and Jan 4 fires are 61 sr and 49 sr, respectively. In the remaining periods, and for the background, we used 50 sr.

Please check the papers Ohneiser et al., ACP, 2020 and Ohneiser et al., ACP, 2022. These papers may serve as reference for all the satellite observations (CALIOP, OMPS, AOD).

We computed the effective lidar ratios of individual smoke layers as in Martinsson et al. (2022). In that paper we compared our CALIOP data to OMPS-LP. It showed good agreement in periods when the OMPS-LP sensor could provide quantitative data (see for example Figure 5 in Martinsson et al. (2022)). We are somewhat confused by this comment. We have all expertize required to process and analyze satellite derived data on aerosol. We have more than a decade of experience with producing in-house satellite products using CALIOP level 1 data (the least processed version provided by NASA). For example; We reported that previous studies had underestimated the climate impact of explosive volcanic eruptions by neglecting the aerosol load in the LMS (Andersson et al., 2015, Nature Communications); We studied a decade of volcanic impact on the climate by correcting CALIOP lidar data for light attenuation caused by stratospheric aerosol particles (Friberg et al., 2018, ACP).; More recently, we developed methods for handling CALIOP data of stratospheric smoke layers (Martinsson et al., 2022, ACP).

You argue that the life-halftime is 10 days because the smoke particles dissolve. How could Ohneiser et al., ACP, 2022 observe the smoke then for 1-2 years after the emission? Please comment on that.

The organic part of the aerosol (90% of the nearfield aerosol) had such short life-time. We argue that organic aerosol is susceptible to photo-oxidation (as predicted by modelling). This portion of the aerosol has a half-life of 10 days. Aerosol stripped of organics remain in the stratosphere throughout the period studied in our work, i.e. during one year. We describe this more clearly in the updated manuscript, and also cite the finding by Ohneiser et al. (2022) suggested by the reviewer.

20°-80°S mean, this is from the tropics to the polar region. What about 30°-70°S? That can be better compared with the reference lidar observations above Punta Arenas at 53°S.

We attempt to study the "total" stratospheric impact of the smoke by capturing regions impacted by smoke. This way we can study the evolution of the smoke without impact of latitudinal mixing that otherwise would have resulted in transport induced variability of the AOD. This enables the study of the AOD decay in Figure 10.

Referencing seems to be arbitrary, especially in the Introduction. A good overview: What is already available (regarding this record-breaking event) together with appropriate references would be helpful! Which gaps are left and filled by this paper?

We aim at explaining the background needed to understand the importance of our study. We moved one of the paragraphs from section 3.8 to the Introduction section, and added additional references on smoke transport to the stratosphere. We were not aware of the simulations of radiation heating and self-lofting performed by Ohneiser et al., 2023 at the time of writing. That study fits well with the topic of our paper regarding cross-TP transport of smoke, and has been added to the Introduction section.

We tried to contribute to filling (at least) two knowledge gaps, i.e. a) how the smoke was transported to the stratosphere, and b) how the smoke impacts the stratospheric aerosol load in the short- and long-term.

- a) We agree that cross-TP transport could be introduced to the reader already in the Introduction section (and not only in the discussions section). We therefore added information and references to troposphere-stratosphere transport in the Introduction section. Here we also added the findings from Ohneiser et al. (2023).
- b) We find rapid decay of fresh smoke in the first month after the 1st fire and compute the aerosols life-time, verifying our previous findings in Martinsson et al. (2022) on smoke from the Aug 2017 North-American fires. Necessary background is already explained on this matter in the Introduction section.

**Specific concerns:**

9 lofted We changed this according to the reviewers suggestion

12 It was so inhomogeneous, how could one determine half-life? Decay is also a function of horizontal meridional dispersion.

We determined the half-life by two methods: 1.) directly from the AOD, and 2.) by studying aerosol - water vapor composition of selected smoke layers, both leading to a half-life of 10 days. Studying the smoke half-life directly from the AOD (Method #1), we tried to encircle the entire region impacted by smoke. Here we used data from all longitudes, latitudes from 20-80 degrS, and altitude from the tropopause to 35 km. With Method #2, we studied the evolution of the aerosol relative to the water vapor composition of individual smoke layers.

Both methods lead to a half-life of 10 days. This is the same half-life as in our recent study on stratospheric smoke from the North-American fires in August 2017.

21 Source We added a reference to the review on stratospheric aerosol by Kremser et al. (2016).

22 define extratropics We changed to midlatitudes and polar regions to be more specific.

34 References We have added the references.

**41 eruptions without `We changed this according to the reviewers suggestion**

51-56 Aging does not just remove/decrease smoke mass. How is that linked to the findings in Ohneiser et al., 2022: 20 months measurements, slow decay or even Canada, Siberia 5-8 months residence time. What about dispersion? Please add references.

In that section we refer to the smoke evolution in the first few months after the smoke injections to the stratosphere after the Aug 2017 North-American fires based on our recent study Martinsson et al. (2022). We claim that 80-90% of the smoke is lost in the first few months. The remaining aerosol stayed in the stratosphere for much longer. We have added a sentence stating that the remaining aerosol stayed in the stratosphere for a year (Martinsson et al., 2022) or even more than one year (Ohneiser et al. 2022).

73 kilometers We changed this according to the reviewers suggestion

80 brackets away around Martinson source We changed this according to the reviewers suggestion

80 define SH We have defined it now. Thanks' for pointing this out.

104 show instead of shows We changed this according to the reviewers suggestion

170-171 AOD CALIOP: What lidar ratios are used? Background and Calbuco and Australian fires. Please compare all these also with ACP, 2020 and Ohneiser et al., ACP, 2022.

We computed the effective lidar ratios of individual smoke layers using the methods described in Martinsson et al. (2022). The mean values used for smoke from the Dec 29 fires and Jan 4 fires are 61 sr and 49 sr, respectively. In the remaining periods. For the background and for volcanic aerosol we used 50 sr.

216 Figure 2 instead of Fig 2 We changed this according to the reviewer's suggestion

226 Decreasing depolarization ratio: put it into context with the findings in Baars et al. 2019 (for example) who found decreasing depolarization ratios after stratospheric smoke injection

Our results are not in contradiction with Baars et al., (2019) since we studied different periods. We studied the smoke evolution over the first 2 months, whereas they studied the evolution from the first 2 months to the first half a year after the fires. Decreasing particle depolarization ratio is expected after the initial depletion of organics, due to mixing with the stratospheric background aerosol which is dominated by spherical particle constituents, i.e. sulfuric acid and water. This effect should be more pronounced once the initial very rapid depletion of organics has occurred, i.e. after 1-2 months in the case of these fires. Hence, our findings of an initial increase in the particle depolarization ratio does not contradict the findings by Baars et al. (2019), but add information on the smoke aerosol evolution during its first 1-2 months in the stratosphere. We have added a reference to Baars et al., (2019) in section 3.4.

Baars et al. (2019) studied smoke from the Aug 2017 North-American fires, and write that the particle depolarization in the first two months were higher than during months 3-4 and 5-6 (0.15-0.25 in Aug-Sep, 0.05-0.10 in Oct-Nov, and <0.05 in Dec-Jan).

For the Australian fires, we find increasing particle depolarization ratio for both fires during the first 4 weeks for both the Dec 29 and Jan 4 fires. After 4 weeks the particle depolarization ratios reached constant values for smoke from the Dec 29 fires. Similarly, we found increasing particle depolarization ratios for smoke layers during the first 4 weeks after the Aug 2017 North-American fires (Martinsson et al., 2022). (For the Australian Jan 4 fires we could not continue to follow the smoke for longer than one month due to faint layers).

233-244 Same as earlier comments: What about sedimentation, what about dispersion, why can the smoke be observed for such a long time, when you write that half-life is 10 days?

This question is thoroughly answered in Martinsson et al., (2022), the first three paragraphs of the Discussion section. We have considered transport out of the stratosphere, sedimentation, cloud formation, and hygroscopic growth as explanations of the decline and found that loss of material from the particles by photolysis is the plausible explanation for the decline (Martinsson et al., 2022). This is also supported by laboratory experiments (Molina et al., 2004; Sareen et al., 2013) and modeling (Hodzic et al., 2015). We added this information in Section 3.6 together with the references.

The majority of the aerosol had such short life-time. We argue that organic aerosol is susceptible to photo-oxidation (as predicted by modelling). This portion of the aerosol has a half-life of 10 days. Aerosol stripped of organics remain in the stratosphere for the entire period studied.

239-244 typically, the findings tend to a decreased BC fraction with time as the coating with organics increases with time. This is in contradiction with your findings please comment on that.

In the troposphere, organic VOCs oxidizes to less volatile species and form SOA, within hours. SOA is not thermodynamically stable and simulations indicate that it becomes photo-oxidized, depleting the organic aerosol. This process is difficult to study in the troposphere, due to dominating wet-scavenging.

273 you could add Ohneiser et al. 2023 in that context (Ohneiser, K., Ansmann, A., Witthuhn, J., Deneke, H., Chudnovsky, A., Walter, G., and Senf, F.: Self-lofting of wildfire smoke in the troposphere and stratosphere: simulations and space lidar observations, Atmos. Chem. Phys., 23, 2901–2925, https://doi.org/10.5194/acp-23-2901-2023, 2023.)

We added the following sentence to the manuscript: "...Most recently, Ohneiser (2023) computed heating and lofting rates for light-absorbing smoke layers throughout the troposphere and the lower stratosphere. Their studies indicate that smoke layers can rise from the UT to the stratosphere via radiation heating..."

282-283 Rieger et al., GRL, 2021 show the opposite. Most of the smoke went south to 70-80°S. Why should the efficient transport pathway be to the north? In your Fig. 4, also most of the smoke is located south of 60°S.

The UVAI Figure (old Figure 11) shows the first weeks after the fire events, and is used here to discuss the increases in UVAI in the days after the PyroCb events. The instrument used in Rieger et al. (OMPS-LP) does not provide quantitative data during this period of dense smoke layers. We agree that our discussion on this topic was somewhat confusing. We have changed the text to better explain how we interpret the UVAI: "...the temporal evolution in the UVAI (Fig. 10a) indicates that most of the smoke remained north of 40°S in the days following each fire event when most of the UVAI was generated..."

312 Aging works also the other way around: Condensation of gases onto the particles, therefore they can have a long lifetime.

As stated above, SOA formation occurs on short time-scales. It is an intermediate, not thermodynamically stable. Simulations indicate that it becomes photo-oxidized, depleting the organic aerosol. This process is difficult to study in the troposphere, due to dominating wet-scavenging.

365-373 These three literature entries are not included in alphabetical order Thank you for pointing this out. We have put them in alphabetical order.

Fig. 1: Too busy, no text in the figure readable, too many subfigures, no legend, no continent boundaries visible, no latitude and longitude range visible

We agree that the Figure was difficult to interpret. We have added a more extensive illustration of the separation of smoke from the two fires as a supplementary (Figure S1).

Fig. 2: X axis text missing, too many figures, too small text, use (a), (b), (c)... in the figure to be able to refer to different parts of the figure. The figure organization is very confusing. Always show the same right below each other and not all the subfigures in different sizes. Please use less subfigures if not every figure is important for the paper.

We have updated this figure based on the reviewer's suggestion. It is now less busy, with fewer CALIOP scenes and with indexing of the subfigures (a ,b ,c ...). To give a more complete picture of the smoke layers from the two fire events, and to further illustrate the differences in the smoke's depolarization ratio from the fires, we also added more than one month (December 31 to February 4) of **CALIOP curtains** of attenuated backscattering, and depolarization ratio for smoke layers (Figure S2-S43).

Fig. 3: Same as Fig. 2, too small, too confusing, too much.

We have updated this figure. It is now less busy. To give a more complete picture of the smoke layers from the two fire events, and to further illustrate the differences in the smoke's depolarization ratio from the fires, we also added CALIOP curtains of smoke layers for December 31 to February 4.

Fig. 5: What is the orange point? What about self-lofting impact?

The orange dot indicates the location and time of the 1st event. We have added this information to the figure caption. We illustrate the stratosphere with three layers that captures the "ordinary" stratospheric transport. The impact of self-lofting is shown in subfigure a), where it takes time for the smoke to enter above 470K (the deep BD-branch).

Fig. 6: What was the used Calbuco lidar ratio and Australian fire lidar ratio? 30-70°S would be better.

As mentioned above, we computed the effective lidar ratios on individual smoke layers using the methods described in Martinsson et al. (2022). The mean values for smoke from the 1st and 2nd event are 61 sr and 49 sr, respectively.

Smoke was present in the region 20-30°S. We used 20-80°S in an attempt to encircle the entire region impacted by stratospheric smoke from the Australian fires.

Fig. 7: How are these results in agreement with the only reference dataset in Ohneiser et al., ACP, 2020 and Ohneiser et al., ACP, 2022? There could be saturation effects.

We do not understand the comment on saturation effects. We corrected the CALIOP data for attenuation by molecules (including ozone) and particles using the methods described in Martinsson et al. (2022).

Fig. 10: What about using latitude subregions: 30-40°S, 40-50°S, 50-60°S, 60-70°S?

Using a narrow latitude range would conflict with the purposes of the present manuscript. We tried to capture the region impacted by the smoke. In latitude-subregion-graphs most of the variability comes from transport in and out of the latitude regions. We therefore tried to encircle the regions impacted by smoke and study the "total" stratospheric impact of the smoke.

Review of Friberg et al., "Short and long-term stratospheric impact of smoke from the 2019/2020 Australian wildfires." (Hereafter "F23").

**Reviewer: Mike Fromm**

**Overview**

F23 present a new satellite-observation-based analysis of the Australia New Year (ANY) 2019/20 fire/pyroCb event. Their aim is to quantify the stratospheric smoke burden and evolution, resolving the two separate contributions from the December and January phases of ANY. There are three distinctive and new contributions herein. 1. determination of smoke AOD decay via photolysis, 2. non-pyroCb-pathway stratospheric pollution by the January-phase plume, yet 3. dominant influence of the January-phase smoke on the overall ANY stratospheric perturbation. In this pursuit, F23 employ aerosol data from nadir-viewing OMPS absorbing aerosol index, limb-view OMPS-LP aerosol extinction, CALIOP backscatter, and H2O data from MLS.

This is a pursuit worthy of study and natural for ACP. The ANY case, individually and by comparison to other stratospheric smoke and volcanic events, is still imperfectly understood and the subject of varying accounts regarding transport pathway, physical evolution, composition, and radiative impact. This study intends to reduce the uncertainties in at least two important ways.

However, F23 have taken on a very complicated scenario, involving continuous smoke generation in southeast Australia in late 2019 and early 2020, punctuated by episodes of pyrocumulus (pyroCu) and pyrocumulonimbus (pyroCb) activity, in a dynamical meteorological setting. I was not convinced by F23's core new finding—that the January-phase smoke was injected only into the troposphere and didn't rise into the stratosphere until ten days post event. F23 build their case with a subset of their data items that under-samples the downstream smoke to the detriment of accurate plume-height and transport-pathway characterization. F23's analysis of aerosol-layer stratification in their illustrations is vague. In addition, their considerable reliance on OMPS UVAI maps (Figure 1) to aid in plume distinction as the smoke blends together is problematic.

We used UVAI smoke observations to indicate the geographical position of smoke in the stratosphere and upper parts of the troposphere. CALIOP observations were used to track the altitude distributions of the smoke, since the UVAI cannot tell the smoke's position in relation to the tropopause. Here, we used mostly nighttime data due to its much higher signal to noise ratio. The UVAI is daytime data, resulting in a temporal displacement. We cover this using the wind patterns at the altitudes of the smoke (as indicated by CALIOP). Furthermore, the depolarization ratios were much higher for smoke from the 1st fire compared with that of the 2nd, which is clearly shown in the supplementary and in the old Fig 2, 3, 8, and 9. Hence, misclassifying smoke would have resulted in overlapping particle depolarization ratios in Fig. 8, which we do not see. We are therefore confident in the separation of smoke from the two fire events.

F23 comes on the heels of two other papers attributing ANY's unprecedented mass of stratospheric smoke to a non-pyroCb pathway. Hirsch & Koren (2021) found no pyroCbs, only random oceanic thunderstorms to be the pathway. Magaritz-Ronen, & Raveh-Rubin (2021) concluded that pyroCbs did not suffice, and that a single Pacific-Ocean synoptic-scale cyclone on ~2 January performed the task. Now F23 attribute the preponderance of ANY stratospheric smoke to tropospheric self-lofting as the pathway while briefly dismissing the above two explanations. Given that Peterson et al. (2021) demonstrated clusters of pyroCbs in both December and January that penetrated the tropopause, and unprecedented UVAI plumes immediately following each impulse, is it F23's position that neither the pyroCb nor the cyclone nor thunderstorm pathway can explain the long-lasting stratospheric smoke observations? If so, their analysis of satellite-data, transport, and lofting needs to be much more exacting than what is presented herein.

These previous studies were based on the initial phase (first days) after these intense fires. Peterson et al, 2021 studied the PyroCbs role as transport paths to the stratosphere. Our results are mostly in agreement with their work. We also see PyroCb impact on the stratosphere after the Dec 29 event, and also after the Jan 4 event. We have made this more clear in the revised manuscript.

CALIOP shows only a small immediate impact from the Jan 4 fires. Most of the smoke clearly entered the UT, but not the stratosphere as pointed out also by Peterson et al., (2021). The UT smoke remained in the UT for a week or more. This is clearly shown in CALIOP curtain plots (see Supplement). What is also evident in these plots is a gradual transport of smoke from the UT into the stratosphere, which is illustrated in the new figures added to the revised manuscript. We cannot find evidence that large amounts of smoke were injected to the stratosphere directly by PyroCbs during the 2nd event. This is simply not in line with our observations. Instead, the data points to later transport to the stratosphere.

Smoke from the 2nd fire had lower particle depolarization ratios than did smoke from the 1st fires. This is clearly shown in the curtain plots in Figure 2 and 3, and in Figure 8 and 9 (now Figure 1, 2, 7, 8). This difference is evident already at the first smoke observations and the difference continues as the

particle depolarizations increase over time. Figure 9 (now Figure 8) and the new Figure 11 show the addition of smoke to the stratosphere more than a week later than the PyroCb formations after the 2nd event (Jan 4). This is visible in all three subfigures in Figure 9. It is more than an indication of a later smoke addition to the stratosphere.

The figures provided by the Reviewer indicate that only little smoke entered the stratosphere directly via PyroCbs from the 2nd fire, although it is difficult to interpret limb-oriented measurements in dense layers. The CALIOP curtain plot from Jan 8 show a small contribution. The OMPS-LP curtain plots are difficult to interpret due to the instrument's limited vertical resolution (~2 km), but they indicate only small contribution of smoke. Furthermore, the OMPS-LP cannot discriminate smoke from ice clouds. Hence, the smoke layers in the UT cannot be resolved with OMPS-LP. The figures provided by the reviewer do not show the UT smoke due to instrument limitations.

My assessment is that if F23 can mount a convincing argument of a diabatic troposphere-tostratosphere pathway for the January-phase plume, the balance of material could become defensible and thus merit publication. But as will be elucidated in my report, F23's analysis of the December and January ANY smoke impulses is unclearly developed and plausibly incorrect. Hence, I recommend major revisions of this manuscript after considering all the concerns I list below.

Peterson et al. (2021) showed that pyroCbs brought smoke into the stratosphere mostly Dec. 29 - 30, and to a lesser extent, Jan. 4, because the elevated tropopause in the latter case resulted in termination of most of the pyroCbs rise in the UT. Their study of the instant events, the pyroCb formations, set the foundations for the developments of the Australian fires. Our evaluation shows that the transport of smoke into the stratosphere continued long after the events studied by Peterson et al. (2021). Just as these authors we find from CALIOP that large amounts of smoke were injected into the UT from the Jan. 4 fires. We see a gradual transport of smoke into the stratosphere from strong pyroCbs that reached the UT, primarily during Jan. 4. The basic mechanism is PyroCb formation, where a delay of transport into the stratosphere is caused by the high tropopause during the Jan. 4 PyroCbs.

The new Figure 11 show that more and more smoke entered the stratosphere over the first weeks after the 2nd event. They also show that the potential temperature for these layers increased over time. The conclusion we draw from this is that radiative heating played a role in transporting smoke from the troposphere to the stratosphere. Quantifying each mechanisms impact on this additional cross-TP transport is outside the scope of this article, but we are happy to collaborate with others on the matter in future studies.

**General Concerns**

A major concern of mine is that F23 discounted a crucial element of the pyroCb "smokestack" phenomenon. The ANY event, like all pyroCb cases, involves the direct injection of smoke, ice, and biomass-burning gases to the top of the pyroconvective column. Like all convective exhaust, these materials disburse at column-top altitudes. These injection heights are quantified in the case of ANY and several other pyroCb events (such as Black Saturday, PNE, Chisholm, etc.) by weather-radar reflectivity and/or infrared brightness temperature image data. In the case of each, it is known that the column-top estimates are generally conservative, i.e. low biased. This has been documented (Fromm et al., 2021, https://doi.org/10.1029/2021JD034928 and references therein). Hence the source term for any such pyroCb event is reliably known to the extent that the pyroCb clouds are thus characterized. In the case of ANY, Peterson et al. (2021) provided an exhaustive accounting of all of the December- and January-phase source terms. Both the December and January phases embodied plumes that topped out in the LMS. Detection of those plumes post-pyroCb are then subject to spotty sampling by any and all satellite instruments. It is regularly the case that pyroCb-plume case studies involve these imperfect satellite data items convolved with various time lags between injection and ideal sampling. Because these fire events also involve somewhat continuous emissions throughout a wide range of vertical transport that the downstream plume picture is embodied by thick and thin smoke plumes from the surface to the top most injection heights. Incomplete/imperfect sampling of these plumes can be suggestive of a multitude of pathways and processes. In the case of ANY, we can state with certainty that the December and January pyroCbs created smoke plumes that topped out above the tropopause ambient on the active dates. . In the case of the 4 January pyroCbs, echotops exceeded 16 km (and  $\Theta$  exceeding 380 K). Consistent plume heights were observable and traceable to these pyroCbs, 2-3 days post pyroCb. But even if such downstream sampling was non-existent or woefully incomplete, it wouldn't change the underlying veracity of the "smokestack" pathway and endpoint. Therefore, the challenge for identifying any other contributing pathway to the stratosphere in the case where direct injection is clearly established is thereby heightened. F23's Figure 2 and 3 appear to show confirmation that both December and January ANY smoke matches the smokestack expectation. Additional support for this end-to-end connection is provided in the supplement at the bottom of this review. F23 are encouraged to weigh this argument and evidence, and consider how it informs their approach.

PyroCbs reached the stratosphere on Jan 4-5 (we do not disagree with Peterson et al., (2021) on this). The cloud-top altitude indeed shows the maximum altitude of the cloud, but do not tell the vertical aerosol distribution. CALIOP shows that large amounts of smoke was injected to the upper troposphere. The presence of smoke in the stratosphere increases over the following week. This is

clearly shown in our figures, e.g. the new Figure 11 added to the manuscript as well as in Figure 9 (now Figure 8). The new Figure 11 show that more and more smoke entered the stratosphere over the first weeks after the 2nd fire. It also shows that the potential temperature for these layers increased over time. Our study adds information on the fate of the smoke that has been little studied in previous work.

As mentioned above, our evaluation shows that the transport of smoke into the stratosphere continued long after the events studied by Peterson et al. (2021). Just as these authors we find from CALIOP that large amounts of smoke were injected into the UT from the Jan. 4 fires. We see a gradual transport of smoke into the stratosphere from strong PyroCbs that reached the UT, primarily during Jan. 4. The basic mechanism is PyroCb formation.

Figures 1 (daily UVAI maps for ~5 weeks), 2 and 3 (ensembles of CALIOP curtains) are used to make the case that the December-phase of smoke injections immediately reached the lowermost stratosphere (LMS) but the January-phase smoke did not. As one would expect, the smoke emissions over the weeklong pyroCb event generated plumes that were straightforward to separate for a brief period, then became indistinguishable based on the UVAI alone. F23 blended the UVAI data with selected nighttime CALIOP curtains to attribute a portion to December and the balance to January. They discuss how they evaluated plume transport to connect these elements with the two phases, but that transport was only inferred via the day-to-day change in the UVAI plume positions. I could not find any additional tool for source-receptor connection such as Lagrangian trajectories. By its nature, the UVAI depends largely on AOD and altitude such that a low, dense layer's UVAI could be indistinguishable from a high, optically thin layer. It is obvious from F23's Figure 2, 3 that multiple layers, from the free troposphere to the LS, were the rule downstream of ANY. The flat color gradation used in the UVAI maps limits any meaningful discernment of low/high, dense/thin features. And as mentioned above, after the first week of January, the previously distinct UVAI plumes become inseparable. F23 recognize this and qualify their analysis with their selection of CALIOP curtains matched to the UVAI features. However, every CALIOP curtain used by F23 is from a nighttime orbit segment. These are systematically ½ day offset from the UVAI-measurement time. This might be OK if there is only one layer at stake and that layer is governed by very light winds. But given the true complexity of the smoke layering and natural wind-shear involvement, the association of the CALIOP features with the UVAI is wholly uncertain.

We used mostly night data due to its higher signal to noise ratio for this separation, and connected CALIOP to the UVAI maps using data on wind patterns. As stated above, possible misclassification

would have showed up in the particle depolarization. Instead, the depolarization ratios differ markedly. Figure 2, 3, 8, and 9 (now Figure 1, 2, 7, 8) show that the smoke was separated both in altitude and in depolarization.

F23 might have invoked other imagery-based retrievals available at night, such as IR-based CO, but they did not.

We preferred to use satellite products that we are familiar with. The 60 m vertical resolution of CALIOP (at the tropopause) is unmatched by any other instrument.

F23 might have invoked daytime CALIOP curtains, but they did not.

We used nighttime data since it has far higher signal to noise ratio and (as pointed out above) misclassification would have shown up in the particle depolarization ratios.

Hence the reader is inadequately informed about the true association of the CALIOP and UVAI features. On its merits, this complicated ANY smoke event requires a more rigorous and precise accounting between the CALIOP/UVAI and the two ANY injection phases.

We disagree. Previous satellite based studies did not perform thorough analysis on particle optical properties in the first month after these fires. We are to our knowledge the first group that performed such analysis. **The particle properties verifies that we have classified the smoke successfully.**

For defensible reasons, F23 do not employ OMPS-LP aerosol extinction data as they do CALIOP for the assessment of the nascent plume altitude. However, this may have been a missed opportunity, given the limb-view data's coincidence with the OMPS-NM UVAI. While acknowledging F23's cited concerns about OMPS-LP utility in the presence of optically thick plumes, its natural combination with the UVAI can and does allow for a confident characterization of LMS plume-top. Moreover, it will be shown that such a combination reveals a finding of the LMS position of the young January-phase plume days before F23's conclusion.

OMPS-LP shows that some smoke from the 2nd event fires entered the stratosphere on Jan 4-5, and so does CALIOP. We do not disagree on this.

However, telling where the plume-top is does not answer the question on how large impact the plume has on the stratospheric aerosol load/AOD. OMPS-LP does not show how much smoke it was since the aerosol extinction coefficients cannot be quantified, and its relatively limited vertical resolution (2 km) add to the difficulties of using its data in cases where the smoke is positioned close to the TP. F23's main analysis leading them to conclude that the January-phase pyroCb smoke was almost totally relegated to the troposphere (until it diabatically lofted across the tropopause ~13 January) exploits the CALIOP curtains shown in Figure 3 and selected point observations in Figure 8.

The zonal means in Figure 9 (now Figure 8) show evidence of additional smoke transport to the stratosphere more than a week after the Jan 4 PyroCbs. Furthermore, the new Figure 11 show that more and more smoke entered the stratosphere over the first weeks after the 2nd fire. It also shows that the potential temperature for these layers increased over time. The conclusion we draw from this is that radiative heating played a role in transporting smoke from the troposphere to the stratosphere.

Figure 3 has 12 panels, curtains from 5-10 Jan. Most panels show dense smoke straddling or above the tropopause.

The figures show weak signals from stratospheric smoke layers and optically dense tropospheric smoke layers. Yes, there were a direct impact on the stratosphere, but much more smoke lingered below the TP in the week(s) following the Jan 4 fires than what initially entered the stratosphere.

Presumably F23 do not attribute these to 4 January. But it is unclear how they interpret Figure 3. The text discussion calls out Figure 3 but does not offer any in-depth explanation of the various aerosol features. On its face then, Figure 3 seems to contradict F23's premise. A sufficiently detailed discussion, commensurate with the many Figure 3 panels and the multitude of aerosol layers therein, is essential. By the way, this recommendation also applies to Figure 2 and attendant analysis.

We understand that the discussions on Figure 2 and 3 were to brief. We have revised these figures, and added a supplementary file which curtains plots of the smoke layers, extending to Feb 4. We have also added an additional illustration (new Figure 11), showing the 2nd fire's smoke layer's position relative to the tropopause. That graph shows that more and more smoke entered the stratosphere over the first weeks after the 2nd fire.

F23 exploit CALIOP depolarization ratio in an interpretation of particle morphology. Two separate threads are presented. 1. They evaluate a temporal increase of depolarization in the stratospheric smoke, in combination with AOD decay, as evidence of photolysis-imposed loss of organic mass fraction. 2. Small depolarization of the January-phase tropospheric smoke as a sign of aging in a humid environment with a transformation to more spherical particles.

These questions are answered in the following paragraphs.

Regarding point 1, two questions arise. First, if the particles are losing mass, one might expect to see a change in CALIOP color ratio commensurate with particle-size reduction. F23 introduce CALIOP attenuated color ratio in Figures 2 and 3, so it is natural to ask if they analyzed the temporal variation of that quantity in relation to depolarization ratio. If so, those results would be to F23's advantage to report. If not, it is reasonable to suggest that F23 perform this analysis to more fully support their line of argument.

It is unfortunately not possible to use the color ratios of the entire smoke clouds. CALIOP data must be corrected for the strong attenuation of the lidar beam in the dense smoke clouds. It is not possible to use the attenuated (uncorrected) data for color ratios, since the two wavelengths become attenuated to different degree. This attenuation correction is only possible for the shorter wavelength (532 nm).

Also, Baars et al. (2019; https://doi.org/10.5194/acp-19-15183-2019) showed that the 2017 Pacific Northwest stratospheric pyroCb plume decreased in depolarization over the course of ~3 months. Might F23 cite that work and comment on the implications of such transformation? Would that be consistent with a resupply of organics or other shell material on the BC core? Is there evidence of CALIOP depolarization ratio decline after the flattening shown in Figure 8?

Of course that can be mentioned in the paper, but on the other hand this always happens. The depolarization ratio decreases as smoke mixes with background air that normally contains spherical particles, regardless of external or internal mixtures of the aerosol material. We study young smoke clouds, whereas Baars et al (2019) compared young clouds with aged, well-mixed clouds, where in the latter case the background aerosol affects the results.

The individual smoke layers became to faint to follow them for months. We do not know if a decline in particle depolarization ratios occurs after the flattening. Particle depolarization ratios are increasing over time for smoke from both the 1st and 2nd fire events, as well as after the Aug 2017 North American fires (Martinsson et al., 2022). Tropospheric SOA (secondary organic aerosol) formation occurs over time-scales of hours in the boundary layer to days in the free troposphere, suggesting that it should not impact the aerosol properties over time scales of weeks or months. The stratospheric conditions may be different though. Furthermore, our results are not in contradiction with Baars et al., (2019) since we studied different periods. We studied the smoke evolution over the first 2 months, whereas they studied the evolution from the first 2 months to the first half a year after the fires. Decreasing particle depolarization ratio is expected after the initial depletion of organics, due to mixing with the stratospheric background aerosol which is dominated by spherical particle constituents, i.e. sulfuric acid and water. This effect should be more pronounced once the initial very rapid depletion of organics has occurred, i.e. after 1-2 months in the case of these fires. Hence, our findings of an initial increase

in the particle depolarization ratio does not contradict the findings by Baars et al. (2019), but add information on the smoke aerosol evolution during its first 1-2 months in the stratosphere. We have added the Baars et al., (2019) finding of decreasing depolarization to the revised manuscript.

Regarding point 2, is it reasonable to speculate on how particles that have a collapsed, nearly spherical BC core would become less spherical during the aging in the stratosphere? What is the evolution of the depolarization ratio of the tropospheric smoke? Is there a perceptible decline in time, consistent with the proposed aging?

We have not made a special study on this subject, probably the relative humidity is an important parameter. Observations with CALIOP from this fire show constantly a lower depolarization ratio for UT smoke. We recall Tandem-DMA measurements where smoke aggregates almost instantly collapse (during the brief transport time between the two DMAs).

A more concrete concern regarding this point is that F23 generalize the ambient condition of the tropospheric January-phase smoke as "humid." While it is indisputable that the stratosphere is dryer than the troposphere, the speculation here about the ambient conditions implies (to me) that the smoke particles are far below the relatively dry UT. Is it F23's expectation that the January plume was largely lower than the UT (as several of the lower, dense layers in Figure 3 are) and thus lofted by several km before entering the stratosphere? See my earlier comment about the need for a much more precise treatment of the features in Figure 3.

The term "humid" refers to the conditions in the upper troposphere, which are more humid than the stratosphere. We have changed the sentence to address this comment. It now says *"more humid tropospheric conditions"*.

We point on the large presence of smoke in the troposphere as the source delayed transport into the stratosphere. Our investigation does not deal with the fate of each and every smoke layer in the troposphere. It is obvious from the numerous CALIOP curtains in Figs 2 and 3 (now Figure 1 and 2), and in the newly added supplementary, that the smoke in the UT has lower depolarization ratio than smoke in the stratosphere.

I was confused by the general treatment of the December and January plume discussion. It appears that F23 are dealing separately with the vortical, rising plume elements and the overall ANY stratospheric smoke. Given that the two compact plume sub-elements represent a minor contribution to the overall AOD of the ANY smoke, the discussion of the global AOD evolution of the two phases seems to be unrelated to the compact-feature evolution. Perhaps I did not catch on to the interplay between the two themes. Please point out to me what I am missing, or otherwise add some discussion material to help the reader appreciate the two seemingly independent themes.

The confinement within the vortex enabled us to follow those individual smoke layers for longer. Smoke layers outside the vortex became too faint to follow for as long (Figure 8, now Figure 7). In Figure 9 (now Figure 8) we show the zonal mean based on all CALIOP nighttime swaths. Both figures reveal a temporal evolution of the particle depolarization ratios. When computing the decay of smoke aerosol we performed one estimate using only data from the isolated cloud (Figure 10a, now Figure 9a), and one estimate based on all data (Figure 10b, zonal mean, now Figure 9b)

F23 claim that the January-phase smoke plume ascended gradually from the troposphere to the stratosphere over the period of 4-14 January. Figure 3, which chronicles the perceived 4 January smoke via CALIOP imagery terminates on 10 January. There are no additional granular CALIOP curtains filling the gap to 14 January. Presumably F23's suggested rise of the smoke to the tropopause would be discernable beyond 10 January (Figure 3). The reader should have the access to the full interpretation of the smoke evolution to be convinced of F23's evidence. The 4-day gap in this timeline stands as a barrier to this understanding. It is essential for F23 to present this entire timeline or explain how the evidence they show in Figure 3 is sufficient to secure their claim.

We have added curtain plots until Feb 4 in the supplementary file. They show the gradual increase in stratospheric smoke from the 2nd fire caused by transport of dense smoke layers across the tropopause. This is also shown in the new figure (Fig 11), where the individual smoke layer's position relative to the tropopause are illustrated.

Section 3.8, "Smoke transport to the stratosphere" is perhaps the most consequential section of this paper. Yet it is populated by weak, confusing, and misleading points. Next, these are shown in bold followed by my reaction in plain text. **"The NA wildfires in 2017 showed that self-lofting by radiative heating of the dense smoke layers caused smoke to rise from the UT into the LMS (e.g. Khaykin et al., 2018; Peterson et al., 2018)."** This is an apparent mischaracterization of these two cited papers. Neither made that argument, or even hinted at it. The diabatic lofting observations they showed all started at stratospheric altitudes. If I missed these points in the two cited papers, please point them out to the reader. We changed "UT" to "Tropopause".

"Ohneiser et al. (2021) suggested self-lofting of smoke from the midtroposphere as cause of extensive smoke layers in the Arctic stratosphere during in the end of 2019 and beginning of 2020." F23 consider this statement might balancing by citing Boone et al. 2022; https://doi.org/10.1029/2022JD036600 ), who categorically refuted Ohneiser's characterization of extensive stratospheric smoke at that time.

We have changed this sentence and added the findings of Boone et al. (2022). It now reads: "…Ohneiser et al. (2021) suggested self-lofting of smoke from the mid-troposphere as cause of extensive aerosol layers in the Arctic stratosphere in the end of 2019 and beginning of 2020. Whether those aerosol layers consisted of sulfate or sulfate-covered smoke particles is under debate (Boone et al., 2022; Knepp et al., 2022)…"

"Hirsch and Koren (2021) argued that smoke injections to the stratosphere may have occurred in the first week of January via cross-tropopause transport by convective clouds south of the fire region (38°S), where the tropopause height is lower. However, the temporal evolution in the UVAI (Fig. 11a) indicates that most of the smoke remained north of 40°S, and only a minor portion was located south of 45°S." But Figures 1 and 2 show that there was abundant smoke south of 45S in late December and the first week of January. This comment is in no way an endorsement of Hirsch & Koren's claim; it is meant to point out that F23 show data that are in apparent direct conflict with this statement.

We do not fully understand this comment, since we have interpreted Figure 11a (now Figure 10a) similar to the reviewer. We do not see evidence of Hirsch & Koren's claim. We have added the following sentence to Section 3.8 to clarify this: "…From the 1st event we do not see evidence of extensive cross-tropopause transport beyond the initial PyroCb caused smoke injections in the CALIOP data…".

The 2nd event fires (Jan 4) positioned dense smoke layers in the mid and upper troposphere (Fig. 3). Figure 3 shows tropopause-level and/or LMS smoke in 9 of the 12 panels. As I commented above, F23 do not examine Figure 3 (or 2) in granular detail or perform an explicit feature-by-feature source attribution. The reader is given no framework on which to match the above statement with the illustrations.

The newly added supplementary file show that most of the smoke was injected below the stratosphere. These regions are the UT and the mid-troposphere, e.g. ranging from the tropopause to many kilometers below the TP. In the newly added figure (Figure 11) we illustrate the 2nd fire's smoke layers position relative to the tropopause (based on the curtain plots provided as supplement).

We see evidence of vertical transport during the following week. What is this evidence? If this is referring to Figure 3, the evidence is not at all clear to me. For the reader to see F23's evidence, a much more detailed analysis of Figure 3 is required.

This is shown by the new figure (Figure 11). Although Figure 9 (now Figure 8) already shows this. It is evident that additional smoke entered that stratosphere more than one week after the PyroCb formation (Jan 4). This impacted all parameters illustrated in Fig 9, i.e. the extinction coefficients, the scattering ratios, and also the particle depolarization ratios.

The generally low depolarization ratios (<0.10) are not indicative of cloud formation. What is the relevance of this statement? Since this is referring to Figure 3, it is apparently meant to compliment F23's argument that this is smoke from 4 January Australia. Moreover, there are several large non-cloud features in Figure 3 that have depolarization ratio >>10%. Hence, this statement requires expansion toward a full characterization of the various scenes provided.

We understand that this may be confusing. We have changed the text (section 3.8) to better describe what we mean, i.e. that those dense features in the CALIOP curtains are smoke and not clouds.

"...we suggest self-lofting by radiation heating and isentropic cross-tropopause transport as the cause of transport to the stratosphere for smoke from the 2nd event, thus following the rising trend in the stratosphere, as demonstrated in Fig. 8b, also in the upper troposphere." "We suggest" implies a considerable amount of uncertainty (as opposed to, e.g. "we find" or "we demonstrated"). Prior to this statement F23 claim to see "evidence" of self-lofting in the troposphere. So, the reader needs to know if their evidence is inconclusive, which would leave F23 in the position of just "suggesting" it. If their evidence is shown to be conclusive, then their presumed position would be that tropospheric self-lofting was demonstrated. Please explain the claimed evidence and discuss their confidence level in the interpretation.

The new figure (Figure 11) and Figure 9 (now Figure 8) show that smoke is added to the stratosphere more than one week after the PyroCb formation from the 2nd event (Jan 4). CALIOP shows no evidence of large immediate stratospheric impact from the 2nd fire, instead it reveals large amounts of smoke in the UT, and that more and more smoke enters the stratosphere over the first weeks after the Jan 4 fires.

This statement also mentions isentropic cross-tropopause transport as a pathway to the stratosphere. They made no prior claim of finding evidence of this for the January-phase smoke. Their only treatment of this pathway was in reference to and partial dismissal of Magaritz-Rohnen & Raveh-Rubin (2021). Hence, it is unclear what justification there is for the suggestiveness of isentropic cross-tropopause transport. Please elaborate, or consider dispensing with this suggestion.

We do not dismiss isentropic cross-TP transport as transport path for smoke to the stratosphere from the 2nd fire. CALIOP curtain plots show that more and more smoke enters the stratosphere over time, but large amounts of smoke from the 2nd fire does not enter the stratosphere as early as Magaritz-Rohnen & Raveh-Rubin (2021) suggested from their trajectory studies. We cannot completely rule out that their suggestion of isentropic cross-TP transport in the first few days after the 2nd fire event (Jan 4) caused smoke to enter the stratosphere, but our new figure (Figure 11) and the zonal means of CALIOP data (old Fig. 9: extinction coefficients, scattering ratios, and particle depolarization) tell that the majority of the smoke entered later. Furthermore, the potential temperature of the smoke layers indicates self-lofting (the new figure).

**Targeted Concerns**

L198-200: "The 2nd fire event occurred on January the 4th, but smoke from this event showed only little immediate stratospheric influence (Fig. 3) in line with observations by Peterson et al. (2021)." What are the observations from Peterson et al. that support this statement?

We refer to their estimate that the 1st fire event gave a larger contribution to the stratospheric aerosol load. We have changed our sentence to clarify what we mean: "…*The 2nd fire event occurred on January the 4th, but smoke from this event showed only little immediate stratospheric influence (Figure 2, S8-S13). Also Peterson et al. (2021) reported much larger stratospheric impact from the 1st fire, based on studies of the fires' immediate impact (2021)…"*

**Section 2.3:** In this introduction of the MLS water vapor, it is called for F23 to cite Schwartz et al. 2020; 10.1029/2020GL090831) who did an in-depth analysis of the ANY MLS H2O data.

We did not cite Schwartz et al. (2020) since the focus of our study is the aerosol.

**Figure 1:** How is the AI color scaled? What are the ascending orbits? Why are they shown? All the panels have grainy resolution. It is very difficult to discern all the additional layer features.

We agree that the resolution was low. We have added this information in the supplementary to better illustrate the transport of smoke in the stratosphere. The supplementary file shows the UVAI maps

together with CALIOP curtain plots of smoke from the two fire events. (The orbits shown in the supplementary are from CALIOP). UVAI: Low threshold: 0.75-1; max 50 (red); typical max in fig 15 - 20 (clear yellow).

**Figure 1**. I was not able to find a thorough description of how the green-line demarcation between December and January phases is justified. Please explain in a way that applies to every figure panel. We found the approximate separation line by combining the horizontal information (UVAI) with vertical information (CALIOP curtains) and wind directions. Here, we aim at separating CALIOP data from the two groups. It is worth noting that both the smoke layers altitudes and their depolarization ratios differed markedly adding robustness to this method. These differences in altitude and depolarization ratio is seen in Figure 7 (old Figure 8) and in the newly added supplementary.

**Figures 2 and 3:** The 532 nm attenuated backscatter panels all have the meteorology-data overlay. These are not described in the captions or main text. Moreover, many of the panels are cropped such that the isoline labels are not displayed. If the meteorology data are necessary, they should be appropriately labeled and exploited in the main text. If they are non-essential, please consider reducing clutter by showing just the lidar data.

We have modified these figures based on the comments from both reviewers. We wish to keep the isolines to make the figures more similar to the supplementary. The information is now added to the figure captions to those figures, as well as to the supplementary file.

Define "main layer" and "minor layer".

We changed these terms since they were misleading. We now refer to the two categories of smoke layers from the 1st event as 'dense isolated' and 'other'.

**Figure 8b:** The earliest triangle is ~10 days after the fire event. But the earliest triangle on the map is not where the phase-2 plume was on 14 Jan (10 days post event). This point appears to be close, geographically, to the position on or about 7 Jan when Khaykin et al. started following this plume element. What is the date of this earliest triangle? If it is indeed ~7 January, then the triangles in Figure 8b are offset incorrectly. If instead the x-axis is relative to the December phase alone, this should be explained.

Those data are from 10 days after the fire event (14 Jan). Please see the CALIOP curtain plot S24d. Also, we updated the figure, since the color scale did not properly match the smoke data.

**L287-288.** "However, the depolarization ratios during the first week of January does not indicate any frequent cloud formation connected to the smoke layers." Meteorologically, what is meant by "frequent cloud formation?" What are the smoke layers to which F23 are paying attention? The reader has no basis on which to accept this assertion. Please bolster it with a callout to and detailed analysis of Figure 3, or providing an additional figure in support.

We understand that this may be confusing. We have changed the sentence to "...do not indicate cloud formation connected to the smoke layers..."

**Conclusions, L322. "Smoke was injected to the stratosphere from two events."** The first event was attributed by F23 to the December pyroCbs. What was the second stratospheric injection event? If F23 maintain the claim that the January impulse was strictly tropospheric, should this sentence be amended? If the second stratospheric injection event was from the January pyroCbs, this contradicts the arguments that are the basis of F23's findings. Please clarify.

We understand that "injected" sounds dramatic and hints of PyroCbs. We therefore changed to "added" to clarify that we point to addition of smoke to the stratosphere, and not specifically to PyroCbs penetrating the tropopause. As mentioned above, we study both the stratospheric and tropospheric smoke from the 2nd event. The 2nd event PyroCb injected smoke into the upper troposphere and in part to the stratosphere. We find that the UT smoke was transported into the stratosphere over the course of weeks. This further highlights the importance of PyroCbs as smokestacks, since they can not only impact the stratosphere directly via injection of smoke the stratosphere.

**Technical Points**

Figure 2. "Four days CALIOP curtains...": Insert "of" before "CALIOP."

**We changed the figure caption.**

**Figure 2 caption. "The day of the fire...":** What is meant by this? Should it be "day of the pyroCb? Considering that this was a multi-day pyroCb phase, it would be advisable to revise this terminology.

We have changed the figure caption.

**Figure 2 caption. "malfunction":** A description of the malfunction is called for, along with a citation so the reader knows what the anomaly is.

We have changed the figure to make it less busy, according to comment by Reviewer #1. This resulted in a change in the caption too. All those curtain plots are available in the new supplementary file.

**Figure 2, and throughout the document. "29 Dec.":** The first phase of the ANY pyroCb event was a multi-day affair, 29-31 December. Please consider describing this phase in that way.

We describe that the event started on Dec 29, both in the abstract and in the main text, and now also in the caption to the Figure.

Figure 3 caption. "Same as figure...": Missing Figure number.

Thank you for pointing this out. We have added the number.

Figure 9a: A scaling factor for the extinction coefficient is missing.

The lidar ratio was computed based on the methods described in Martinsson et al., (2022). It has a mean value of 61 sr (1st event) and 49 sr (2nd event). For further details, please see answer to comments from Reviewer #1.

Figure 9c: "1.0" on the color bar: Should be "0.10"

Thank you for finding this. We have changed to 0.10.

**3.2 Compared to volcanism:** Please state what is being compared to volcanism in this section heading.

We changed to "Wildfire smoke compared to volcanism"

**Supplement**

---

## Author Comment (AC2)

[revised manuscript text omitted]

**Figure 2̶3̶. Same as F̶figure 1̶, but for smoke layers from the 2ⁿᵈ fire event (2020-01-04). **

[Figure]

**Figure 34 | Latitudinal and vertical distribution of smoke in Dec 2019 – Mar 2020.** CALIOP zonal mean aerosol backscattering coefficients during the first three months after, and one month preceding, the first stratospheric injection. This parameter can be viewed as an optical version of aerosol concentration.

585

[Figure]

**Figure 5 | Latitude and time distribution of the stratospheric aerosol load.** Aerosol optical depth (AOD) in three stratospheric layers and in all layers combined. The upper (a) and mid (b) layers are the deep and shallow branches of Brewer-Dobson circulation, and the lowest layer (c) is the lowermost stratosphere. The orange dot shows the approximate latitude and the time of the 1st fire.

[Figure]

**Figure 56 | Wildfire impact on the stratospheric aerosol load in the southern extratropics. a) The aerosol optical depth (AOD) from CALIOP in 2013-2020 divided into two-year-steps, illustrating the impact of the Australian wildfire in 2019-2020, together with that from the Calbuco eruption in 2015-2016, and background levels 2013-2014 and 2017-2018. b) the stratospheric AOD from CALIOP during Dec 2019 – Jul 2020 separated into the three layers used in Figure 42. Lines mark 8-day smoothed AOD data, and symbols are daily means of the LMS (yellow), shallow (brown) and upper (pink) BD-branches, and the total stratospheric AOD (grey).**

[Figure]

**Figure 67 | Wildfire impact on stratospheric AODs at three wavelengths in the southern extratropics from CALIOP and OMPS-LP. The stratospheric column above the 380 K isentrope column at 20-80°S.**

[Figure]

605

**Figure 78 | The smoke transport and chemical evolution in the stratosphere. a) Geographical transport of the smoke from the two injection events. b) Ascension of the smoke clouds, and c) temporal evolution of the particle depolarization ratios for the two injection events. All data taken on individual smoke clouds in the stratosphere. Circles mark data taken in the dense isolated cloud from the 1st event fires, diamonds are other data from the 1st fire, and triangles mark smoke data from the 2nd event fires.**

610

[Figure]

[Figure]

[Figure]

**Kommenterad [JF2]:** Updated the tick labels according to comment from Reviewer #2. (1.0 was changed to 0.10)

**Figure 98 | Separation of the smoke from the two phases of the fires. Stratospheric zonal mean (20-80°S) a) Extinction coefficients, b), scattering ratios (SR), and c) depolarization ratios where SR values are higher than 5. The dashed and full lines represents the separation line of smoke for the two phases of the fire, and the minimum altitude used to compute the AODs for the two phases of fires.**

[Figure]

[Figure]

**Kommenterad [JF3]:** We changed the y-axis labels in Figure 9a and b from 'AOD increase' to 'AOD anomaly'.

**Figure 9 | Smoke decay in the dense isolated cloud from the 1st event. a) 8-day running mean of the background subtracted stratospheric zonal mean AOD at 20-80°S above 14 km altitude for the 1st and 2nd phase, respectively. b) Ddaily means of background subtracted AODs for the 1st phase only, and c) smoke data from individual smoke layers (scattering ratios, SR, from CALIOP) normalized with water vapor concentrations (cH2O, from MLS). The exponential fits correspond to a smoke half-life of 10 ± 2 (b) and 10 ± 3 days (c).**

[Figure]

625 **Figure 10. | Evolution of the UV aerosol index (UVAI) during the first weeks after the fire events for a) zonal means and b) meridional means. The locations and dates of the events are marked by orange dots. The lower UVAI range in a) stem from the method of area weighting data.**

[Figure]

630

**Figure 11. | Ascension of 2nd event smoke layers for a) smoke layer position relative to the tropopause (TP), and b) smoke layer potential temperature. Circles mark the layer midpoints, downward and upward pointing triangles mark the layer tops and bases, and the grey line marks the tropopause. Data were retrieved from CALIOP curtain plots (see supplementary for further details), where layers with layer tops below 8km where included in the graph.**

635

**Table 1 | Wildfire and volcanic impact on aerosol optical depths and radiative forcing. The one-year AOD increase in the extratropics (20-80°N/S) after the largest volcanic eruptions and wildfires since 2006 compared to the Australian wildfires.**

| Date | Location | | Event name | 1 y AOD incr. | 1 y AOD incr. CALIOP |
|---|---|---|---|---|---|
| | Lon. | Lat. | | CALIOP | rel. Aus. Fires |
| 2008-08-07 | 176°W | 52°N | Kasatochi | 0.0059 | 63% |
| 2009-06-12 | 153°E | 48°N | Sarychev | 0.0090 | 97% |
| 2011-06-12 | 42°E | 13°N | Nabro | 0.0057 | 61% |
| 2015-04-23 | 73°W | 41°S | Calbuco | 0.0080 | 86% |
| 2017-08-12 | 120-125°W | 49°N | N.Am.Fires | 0.0027 | 29% |
| 2019-06-22 | 153°E | 48°N | Raikoke | 0.0104 | 110% |

| 2019-12-29 - 2020-01-04 | 147-151°E | 34-38°S | Aus. Fires | 0.0093 | - |

**Supplementary to: Short and long-term stratospheric impact of smoke from the 2019/2020 Australian wildfires**

**By:** Johan Friberg, Bengt G. Martinsson, and Moa K. Sporre

**Text S1:**

The figures (S2-S42) were used to track the smoke layers from the Australian fires. The figures consist of UV-Aerosol index (UVAI) images (top figures) with included CALIOP swath paths from the same day. Here, the lower threshold for UVAI was set to 0.75 to exclude aerosol at low altitude. The curtain plots from the CALIOP swaths (attenuation and depolarization) are shown below in each figure. Some days contained too many CALIOP curtain plots with smoke layers to show in a single figure. The plots for those days was divided into two figures. Mainly CALIOP nighttime swaths has been used in the analysis since the daytime data have low signal to noise ratios. Note that the CALIOP nighttime images are not co-occurring with the UVAI data due to temporal displacement between the satellites. Thus, the smoke layers are often placed further east in the UVAI compared to when the CALIOP image was recorded due to westward transport of the fire clouds.

In the figures, blue circles/lines mark aerosol from the $1^{st}$ event in the isolated dense cloud. Red circles/lines mark aerosol from the $1^{st}$ event that did not belong to the isolated cloud. These could not be tracked for as many days as the smoke layers in the dense smoke cloud. Green circles/lines mark aerosol from the $2^{nd}$ event. The CALIOP swaths was used to detemine the altitude of aerosol from the $2^{nd}$ event and to track its movement over time. In the attenuated backscatter figures, the tropopause height in connection to the fire clouds has been determined (white lines). The layer bases could not be determined directly from the attenuated backscattering due to attenuation by smoke aerosol. We therefore determined the layer bases from the depolarization ratio plots. The layer tops (white lines long dashes), layer mid-points (white line) and layer bases (white lines with narrow dashes) are marked in these plots. The orange lines in the plots mark the separation between the smoke layers from the $1^{st}$ and $1^{nd}$ event.

The smoke layers are distinguished and separated from clouds by their depolarization ratios. Smoke layers have depolarization ratios <0.20. Since the UVAI values are influenced by the altitude of the aerosol, a high UVAI do not always result in a strong attenuation signal in the CALIOP data, and vice versa. Some Caliop curtain plots were excluded from the analysis since the aerosol attenuation signals were too weak even though the UVAI indicate presence of smoke aerosol. As the smoke layers are dispersed over time they become more difficult to track and determining their altitude is no longer possible.

[Figure]

Figure S1. Daily OMPS-NM UVAI maps from 2019-12-31 to 2020-02-04. These data are used in connection with vertical information from CALIOP, where we had no meaningful data on 2019-12-29 and no data at all 2019-12-30 from the overpass of the region of elevated UVAI. Ovals indicate positions of individual layers used in Figure 7: Blue indicate the dense isolated smoke from the 1st event fires, red indicate other smoke from the 1st fire, and green indicate smoke from the 2nd event fires. The green line indicates approximate limit between layers from the 1*st* and 2*nd* fire.

[Figure]

Figure S2. For figure description, see Text S1.

[Figure]

Figure S3 . For figure description, see Text S1.

[Figure]

Figure S4. For figure description, see Text S1.

[Figure]

Figure S5 . For figure description, see Text S1.

[Figure]

Figure S6. For figure description, see Text S1.

[Figure]

Figure S7. For figure description, see Text S1.

[Figure]

Figure S8. For figure description, see Text S1.

[Figure]

Figure S9. For figure description, see Text S1.

[Figure]

Figure S10. For figure description, see Text S1.

[Figure]

Figure S11. For figure description, see Text S1.

[Figure]

Figure S12. For figure description, see Text S1.

[Figure]

Figure S13. For figure description, see Text S1.

[Figure]

Figure S14. For figure description, see Text S1.

[Figure]

Figure S15. For figure description, see Text S1.

[Figure]

Figure S16. For figure description, see Text S1.

[Figure]

Figure S17. For figure description, see Text S1.

[Figure]

Figure S18. For figure description, see Text S1.

[Figure]

Figure S19. For figure description, see Text S1.

[Figure]

Figure S20. For figure description, see Text S1.

[Figure]

Figure S21. For figure description, see Text S1.

[Figure]

Figure S22. For figure description, see Text S1.

[Figure]

Figure S23. For figure description, see Text S1.

[Figure]

Figure S24. For figure description, see Text S1.

[Figure]

Figure S25. For figure description, see Text S1.

[Figure]

Figure S26. For figure description, see Text S1.

[Figure]

Figure S27. For figure description, see Text S1.

[Figure]

Figure S28. For figure description, see Text S1.

[Figure]

Figure S29. For figure description, see Text S1.

[Figure]

Figure S30. For figure description, see Text S1.

[Figure]

Figure S31. For figure description, see Text S1.

[Figure]

Figure S32. For figure description, see Text S1.

[Figure]

Figure S33. For figure description, see Text S1.

[Figure]

Figure S34. For figure description, see Text S1.

[Figure]

Figure S35. For figure description, see Text S1.

[Figure]

Figure S36. For figure description, see Text S1.

[Figure]

Figure S37. For figure description, see Text S1.

[Figure]

Figure S38. For figure description, see Text S1.

[Figure]

Figure S39. For figure description, see Text S1.

[Figure]

Figure S40. For figure description, see Text S1.

[Figure]

Figure S41. For figure description, see Text S1.

[Figure]

Figure S42. For figure description, see Text S1.

---

## Referee Report (RR1)

The revised version of the manuscript of Friberg et al., 2023 can almost be published as is.

It is visible that the authors put a lot of effort in the revision process of the manuscript. All concerns and questions were thoroughly addressed.

The authors convincingly argued about their two independent methods to retrieve the short decay time of 10 days and that this finding is not in contradiction with existing literature.

Thanks for improving the literature, introduction, and figures. Now all of the figures are better readable. The font size of latitude and longitude in Figs. 1 and 2 could, however, still be improved.

The used values of 61 sr and 49 sr for the Australian wildfire smoke appear too small for me. It is okay that you use the recommended values in Martinsson et al., 2022, but maybe you could refer to the fact that also other lidar ratios are used in literature for the same smoke plume.

---

## Referee Report (RR2)

Review of revised Friberg et al., "Short and long-term stratospheric impact of smoke from the 2019/2020 Australian wildfires." (Hereafter "F23").

Reviewer: Mike Fromm

F23 are to be commended for the efforts to update and improve the material they presented. Their track-change document is very helpful to the reviewer. These changes reduce, in many ways, the areas of uncertainty and confusion that I harbored regarding the initial submission.

However, these changes were not persuasive in terms of F23's assertion that 1. the January-phase plume had an insignificant initial stratospheric component and 2. that January-phase Australia smoke was transported from the troposphere to the stratosphere. Each of these is discussed in turn, in the next section.

My overarching concern with F23 is that they posit diabatic self-lofting from the troposphere to the stratosphere as a pathway leading to a stratospheric smoke pollution event greater than the direct pyroCb pathway acknowledged as having occurred five days earlier. This is an extraordinary claim, which requires extraordinary evidence in support. Given the acknowledged circumstances (the a priori existence of a major stratospheric smoke plume), this paper must show incontrovertible evidence of a slow, diabatic intrusion to the stratosphere that is as definite as the already published material on the Australia Black Summer pyroCb event. The manuscript does not meet that challenge, in my assessment. The ANY event still calls for explorations such as this. But the complexity of the atmospheric situation is not resolved or clarified by F23's analysis to my reading. Absent that outcome, I cannot recommend this publication. Major changes are called for.

**General Concerns**

Regarding point 1 above, F23's messaging is that their data and interpretation were supported by Peterson et al. (2021). But my assessment is that the data are still contradictory and the Peterson et al. results are mischaracterized. F23 downplay the January-phase stratospheric injection by citing Peterson et al.'s revelation that the December-phase injection mass was 2-8 times greater than that of the December phase. However, the January-phase injection mass was by itself on par with the 2017 Pacific Northwest pyroCb event (PNE), which was at that time unprecedented in the satellite record for stratospheric smoke perturbation. Hence it is reasonable to imagine an Australia event that consisted only of the January phase. The Peterson et al. stratospheric smoke-mass injection could logically have resulted in a lasting plume on par with PNE. Peterson et al. calculated the stratospheric smoke source term by two methods, one based solely on the UVAI. The UVAI maximum, combined with the stratospheric area of the UVAI plume, showed that the January-phase event was on par not only with PNE, but all other historical documented pyroCb events as well.

As useful and advantageous as CALIOP data are for the purpose of characterizing nascent pyroCb plumes, they are—like all other remote sensing data items--subject to the vagueries of

optimal sampling. The nature of sampling and the naturally small footprint of a nascent smoke plume is that they are often missed or incompletely captured with satellite data like CALIOP. My intense experience with the study of the ANY event, no doubt the same as F23's, revealed that the January-phase smoke plume was missed by CALIOP such that ideal plume height and concentration was delayed for several days. There were missing complete orbits or orbit segments on some days, and other days where CALIOP was just unlucky. Our survey of CALIOP showed that CALIOP made some luckier samples of the nascent December plume, as shown in F23's figures. This goes to the heart of why I asked in the first review about daytime CALIOP curtains and OMPS-LP, even conceding the poorer quality of those data. F23 showed in their supplement several daytime CALIOP curtains, implying the recognition of their value toward a more complete characterization of the nascent smoke plume.  The unfortunate lack of perfect CALIOP sampling should not be convolved in the interpretation of plume altitude and concentration. In the case of ANY and PNE, the UVAI-based estimates of the immediate stratospheric presence of smoke are an independent and robust marker of the stratospheric smoke source term. In short, there is a large, built-in uncertainty to the early sampling of CALIOP, even if all the CALIOP data are utilized. Unless F23 contend that the Peterson et al. January-phase stratospheric smoke mass is largely overestimated, the Peterson et al. results stand as an indication that the 4 January pyroCb event impregnated the stratosphere on par with almost all prior pyroCb events. If the veracity of the Peterson et al. estimates are not disputed, the challenge of showing additional smoke entering the stratosphere thereafter is considerable.  F23 are encouraged to consider this argument and make whatever changes are called for in their line of analysis.

On the second point above, F23 made it much clearer in this revision how they accounted for the separation of December and January event smoke layers. Their CALIOP survey in Figures 1, 2, and Supplementary was more complete; their connection to phase 1, phase 2, and "other" was reasonable.  From those data they built the central figures 7 and 11. This made it easier to understand these two important figures. But in my assessment, it did not adequately prove a diabatic pathway from the troposphere to the stratosphere, which is their central claim. Figure 7 contains only stratospheric observations. The CALIOP data assigned either to December or January are wholly or largely tied to two isolated, contained, and circulating smoke plugs followed by Allen et al. (2020 https://doi.org/10.1175/JAS-D-20-0131.1), Kablick et al. (2020), Khaykin et al. (2020), and Schwartz et al. (2020, **https://doi.org/10.1029/2020GL090831**). F23 expressly tied their December-phase data to one of these "isolated" entities. They did not directly tie their January-phase CALIOP data to an isolated smoke entity, but Figure 7a maps out the path of the plume in a way that compared almost perfectly with that of a smoke-vortex element illustrated in the above-cited papers. The important point is that the morphology of both of these smoke entities is wholly achievable by quasi-Lagrangian means due to their confinement and the absence of compromising uncertainties regarding smoke decay. These two contained smoke vortices perceptibly rose diabatically, in comparison to the extra-vortex ANY plumes (as hinted at by F23's "other" CALIOP data points in Figure 7).

F23's expressed and incidental following of these contained smoke vortices stands in contrast to the tropospheric CALIOP data they present. No attempt was expressed to follow these

CALIOP layers materially. F23 clearly outline the difficulty of characterizing tropospheric smoke morphology because of the various, dominant, ubiquitous processes such as wet deposition. Canonical application of these forces to tropospheric smoke would lead to a vertically stratified decay profile that could appear as identical to the evolution of the CALIOP tropospheric smoke observations in Figure 11. For this reason alone, the apparent slope upward in the troposphere in Figure 11 has more than one explanation, unlike the same slopes shown for the stratospheric subset.

The supplement plots by themselves cannot be used to infer upward transport. Static images 24 hours apart allow no such definitive statement. Moreover, I could not discern a systematic rise in the tropospheric smoke from these figures. If indeed there is such a signal, it would be essential for F23 to analyze and defend that scenario.

F23 should be expected to offer other potential explanations for the pattern seen in FIgure 11. Given that wide-scale tropospheric smoke pollution on par with that seen in this case is not uncommon, and other such occurrences were not followed by significant stratospheric pollution, alternate explanations should be given at least equal weight to diabatic lofting.

It is noteworthy that the smoke vortex associated with F23's January-phase CALIOP data was tracked by Schwartz et al. by virtue of MLS stratospheric water vapor enhancements. If this smoke entity had its origin in the troposphere, it would have been subject to the cold trap at the tropopause, a limiting factor on its water vapor content. The fact that all of the ANY contained plume elements shown by Schwartz et al. were defined by water vapor enhancement, and that they all were based in the lowermost stratosphere, indicates a commonality, that being a direct pathway via pyroconvection. If that is a defensible statement, the challenge for F23 is to show that slowly ascending air masses in the troposphere can deliver not only smoke through the cold-point tropopause, but water vapor plumes as well.

Figure 8 is composed of zonal averages. The range from 20-80S would embody tropopause heights generally above 14 km in the northern realm to lower in the southern realm. Since this analysis is done with an absolute altitude scale, it is to be expected that the data below 14 km is a blend of tropospheric and stratospheric aerosol. Since these data almost certainly represent some unknown blend of stratospheric air, it is uncertain as to how to assess these lowermost data points in relation to those above 14 km.

Both phases produced UTLS smoke with equally small depolarization ratio (depol for short). E.g., see the intense stratospheric layer in Figure S3b,c. The fact that the December phase produced such low depols at such high altitude prompts the question as to the true difference between the December and January nascent UTLS plumes' particle-shape populations. Could F23 comment on that?

F23 have convincingly shown that the January-phase stratospheric depol is somewhat less than the December phase. But what does that necessarily say about its origin altitude? Wouldn't we expect the photolytic process to drive both depol populations to the same eventual value? If so, what would account for the difference in the December and January aged plumes at stratospheric altitudes?

Peterson et al. showed that, by number, there were many more pyroCbs in the December phase that injected only to the troposphere than on 4 January. That may or may not have been evident in the CALIOP curtains shown in F23, but it is plainly evident in the full set of CALIOP curtains that there was abundant tropospheric smoke from 29 December onward. Given Peterson et al's accounting, doesn't it seem reasonable that there was much tropospheric smoke from the December phase that was in place by the time of the January phase?

Technically, F23 do not show diabatic transport from below the tropopause. They infer it. The stratospheric fraction of observations can be accepted as reflecting diabatic lofting largely on the strength of prior publications such as Khaykin et al;, Kablick et al. and Schwartz et al., as well as the several papers on the PNE pyroCb event. But the troposphere-to-stratosphere mechanism has still not been proven with observations.

In terms of the flow of argumentation in F23, I found several instances where they assert their conclusion about the diabatic lofting from the troposphere prior to any detailed analysis. In this way they seemed to put the cart before the horse. It is advised that they not only bolster their analysis proving the tropospheric diabatic lofting, but also withhold any conclusions/assertions until the reader sees the proof.

F23, on occasion, refer to the smoke-layer tops in the LMS as "minor" in relation to the bulk of the aerosol layer. This is reasonable from a descriptive perspective, but considering the strong vertical stratification of aerosol lifetime, this may be prejudicial. There is no doubt that typical pyroCb events distribute smoke throughout the troposphere up to the UTLS, leaving what may appear to be a minor portion at the topmost altitudes. But that "minor" part has a much greater potential to last than the eye-catching tropospheric parts. Moreover, sampling by CALIOP may give the strong appearance of the topmost smoke as being small in proportion to lower plumes. A telling example of this is shown in Figures 6 and 7 here:

https://doi.org/10.1029/2021JD034928.  It shows an early view of the PNE smoke plume that captured a small footprint of the pyroCb smoke plume. This "minor" feature represented the most consequential early indication of the smoke that polluted the stratosphere. Hence F23 are asked to reflect on the "minor" indications of the January-phase LMS smoke as early as 6 January illustrated in Figure 11.

F23 responded to my initial review's question about exploiting CALIOP color ratio. They argued that the differential attenuation made that an insurmountable hurdle. However, they did exploit CALIOP color ratio in Martinsson et al. (2022) in a manner I was alluding to. Hence, I still wonder if one could discern systematics in the temporal evolution of the color ratio. Both the

color ratio and the differential attenuation are associated with particle size. There might be qualitative as well as quantitative tactics to assess the temporal changes even while acknowledging the attenuation issue. Since this group met that challenge in Martinsson et al., they are encouraged to do the same here or explicitly address the reason why they avoid that in this work.

Katich et al. (2022, DOI: 10.1126/science.add3101) used in situ aircraft data to develop a "fingerprint" of pyroCb stratospheric smoke, in comparison to non-pyroCb smoke, in terms of an especially large coating thickness around the BC core. F23 may wish to review this paper and comment on any potential conflicts with their hypothesis of photolytic processes reducing OA material mass.

**Targeted Comments/Concerns**

Below, line numbers, figure numbers, and F23 quotes are in **bold**. My reaction is in plain text.

**L30:** The vertical definition of the LMS is not given herein. The term is used sometimes qualitativley, but also used as part of a targeted calculation of AOD. The details of that calculation need to be provided.

**L53-54, "At least 38 PyroCbs injected smoke to the stratosphere during two events…":** Please revisit Peterson et al. and revise this statement. According to Peterson, only a subset of the 38 pyroCb-pulse injections reached the stratosphere.

**L59-62:** Even with F23's revisions, the statements in this paragraph conflate two previous published conclusions under the banner of transport from the troposphere to the stratosphere. Only Ohneiser et al. fits within this pathway. The Peterson and Khaykin papers start the plume ascent in the lowermost stratosphere. Please add the proper nuance.

**L73, "…aerosol stayed in the stratosphere for a year…(Ohneiser et al. (2022).":** As far as I can tell from Ohneiser (2022), they only show PNE smoke for ~8 months. Did I miss it? If not, please provide another citation.

**L125-126, "The depolarization ratios for smoke from the 2nd fire were clearly lower than those for smoke from the 1st fire…":** While some systematic difference is visually apparent, there is a considerable overlap in depol between the two phases. The December phase generated some very low-depol layers above the tropopause, as did the January phase. There are several additional CALIOP curtains attributable to the December phase, not shown here, that reinforce the realization of overlap in the depol ratio between phases. In general, it appears the free tropospheric smoke depol has single-digit depol, stratospheric has decidedly double-digit depol., and tropopause-level smoke has a wide range, as manifested in both phases. Would the authors care to comment on that?

**L125 paragraph and figures called out:** It would be helpful to have marks such as arrows on the figures pointing to features the authors want to highlight to make their point.

**L127:** What property? F23 describe two populations of smoke depol, but not in contrast to other particle types. Please elaborate.

**L231-232, "Peterson et al. (2021) reported much larger stratospheric impact from the 1st fire, based on studies of the fires' immediate impact.":** Yes, but the stratospheric mass from Phase 2 was equivalent to PNE, according to Peterson's Figure 1. So, on its own merits, the Phase 2 plume was a major stratospheric presence.

**L197 and elsewhere:** "elevation" is regularly used to characterize an increase in AOD. This term also denotes changes in altitude. It might be advisable to choose another descriptor of the AOD amplitude change.

**L232-233, "10 days after the PyroCb formations we start to see more stratospheric influence (Figure. 7).":** Figure 7's first January-phase data point is on 14 January, ten days post event. But Figure 2 and especially supplementary figures show January-phase CALIOP curtains dating to 5 January (1 day post event). Moreover, Figure 11 starts on 5 January. But the reader is first introduced to the January-phase smoke by the callout to Figure 7. So, it seems to be misleading to support the above statement by this figure callout. Figure 11 shows stratospheric influence from the January phase being first detected by CALIOP on or about 6 January. The weight of Figure 2, 11, and the supplementary figures indicates that a re-characterization of this sentence is called for.

**L233:** It's not clear what **"stratospheric influence"** means. Figure 7 simply follows two smoke vortices. This doesn't represent the entirety of the smoke plume. Please consider rephrasing this.

**L236, "Over time, more and more smoke…":** This is not obvious from the CALIOP curtains in FIg 1,2, and Supplement. This is a conclusion stated before any proof is given.

**L239, "…rose by at approximately the same rate as…":** Again, Figure 7 simply follows two vortices, one of which was spawned by the January plume. So it is no surprise; the ascent of that vortex has already been documented by Khaylin, Kablick, and Schwartz.

**L257-258, "Some of the smoke from the 1st event reached the UT (Figure. 1) and may have risen later along with smoke from the 2nd event contributing somewhat to the second AOD peak.":** This is putting the cart before the horse. The reader has yet to see any analysis proving tropospheric lofting.

**L259, "Smoke from the 1st event rose markedly in the stratosphere before smoke from the 2nd event entered the stratosphere (Figure 7b).":** This conclusion cannot be drawn from Fig 7. F23 started the plots on 14 January for the January plume. The various CALIOP curtains prove

that there was stratospheric smoke from the Jan phase many days earlier. See also Peterson et al.

**L265, "…the 2nd event that ascended later into the stratosphere.":** At this point the reader has not been shown proof that tropospheric smoke ascended into the stratosphere. The material that has been presented at this point does not perform that task.

**L295-297, "This explains the low depolarization ratios for smoke from the 2nd event…":** See my comments about the copious observations of low-depol. tropospheric smoke in the December plume. Note also the multiple CALIOP observations of low-depol, tropopause-level smoke from both events.

**L326-327, "From the 1st event we do not see evidence of extensive crosstropopause transport beyond the initial PyroCb…":** The reader does not get any information aligned with this conclusion. What analysis did F23 perform to elicit this finding?

**L354-355, "A continuous crosstropopause transport over the course of several weeks also affects the AOD evolution.":** This pathway is taken as a given here. Cross-tropopause transport is not quantified in a manner to support the claim that it occurred over several weeks.

**L356-357, "Our study indicates that smoke from the 2nd event had larger long-term impact…(Figure 9.":** Nowhere, to my reading, did F23 explain how Figure 9a was constructed. Elsewhere in the paper they described tracking the two phases of smoke only out to 4 February. How did the smoke-phase distinction over several months get accomplished?

**Figure 5:** Do F23 wish to opine on the secondary AOD increase in the winter months? This period is not a focal point of the paper but the feature of increased AOD is in stark contrast to the decay signal, thus potentially more consequential than the earlier AOD dip and rise on which F23 focus.

**Figure 7 caption, "…smoke transport and chemical evolution…":** "chemical" is not shown, only optical properties. Chemistry is inferred from simulations, which is discussed in the text. But the figure caption should describe what is shown.

**Figure 9 caption, "Smoke decay in the dense isolated cloud from the 1st event..":** This first sentence is confusing. The figure panel a shows both events. And panel a's description is "zonal means." How are events segregated while calculating zonal means?

**Figure 11.** The fit lines are not described in the caption.

**Supplementary Material, "Some Caliop curtain plots were excluded from the analysis since the aerosol attenuation signals were too weak even though the UVAI indicate presence of smoke aerosol.":** This is unclear. Since there is no one-to-one relation between night CALIOP

and day UVAI, how did F23 determine the weakness threshold used for excluding CALIOP layers?

**Technical Comments**

**Figure S1:** Please separate the left and right columns. The continuous black background makes it difficult to see the east edge of the left column and west edge of the right column.

Throughout the text, use subscript notation in "SO2."

---

## Author Response (AR2)

We have complemented our analysis of the January 4 event and its transport into the stratosphere with daytime CALIOP curtain plots, which are now also included in the supplementary file. The analysis of the daytime images fully support the results we have presented from the nighttime curtain plots. The nighttime and daytime curtain plots together provide several vertical images of the smoke clouds per day.

See our answers to the reviews comments below.

**Reviewer #1**

**Review of revised version of Friberg et al.**

The revised version of the manuscript of Friberg et al., 2023 can almost be published as is.

It is visible that the authors put a lot of effort in the revision process of the manuscript. All concerns and questions were thoroughly addressed.

The authors convincingly argued about their two independent methods to retrieve the short decay time of 10 days and that this finding is not in contradiction with existing literature.

Thanks for improving the literature, introduction, and figures. Now all of the figures are better readable. The font size of latitude and longitude in Figs. 1 and 2 could, however, still be improved.

We have updated these with larger font size for latitude and longitude, according to the suggestion by the reviewer.

The used values of 61 sr and 49 sr for the Australian wildfire smoke appear too small for me. It is okay that you use the recommended values in Martinsson et al., 2022, but maybe you could refer to the fact that also other lidar ratios are used in literature for the same smoke plume.

The lidar ratio we use are effective lidar ratios, i.e., they are affected by multiple scattering. That makes the lidar ratios smaller. The reason is that the scattering is enhanced by the multiple scattering. This means that these two quantities are affected in opposite directions by multiple scattering, causing a cancelation in the product of them (which is used to compute the AOD). Therefore literature data on the lidar ratio should be avoided for measurements, like CALIOP, affected by multiple scattering. This is explained in Martinsson et al., 2022.

**Reviewer #1**

**Reviewer #2**

Review of revised Friberg et al., "Short and long-term stratospheric impact of smoke from the 2019/2020 Australian wildfires." (Hereafter "F23").

Reviewer: Mike Fromm

F23 are to be commended for the efforts to update and improve the material they presented. Their track-change document is very helpful to the reviewer. These changes reduce, in many ways, the areas of uncertainty and confusion that I harbored regarding the initial submission.

However, these changes were not persuasive in terms of F23's assertion that 1. the January-phase plume had an insignificant initial stratospheric component and 2. that January-phase Australia smoke was transported from the troposphere to the stratosphere. Each of these is discussed in turn, in the next section.

My overarching concern with F23 is that they posit diabatic self-lofting from the troposphere to the stratosphere as a pathway leading to a stratospheric smoke pollution event greater than the direct pyroCb pathway acknowledged as having occurred five days earlier. This is an extraordinary claim, which requires extraordinary evidence in support. Given the acknowledged circumstances (the a priori existence of a major stratospheric smoke plume), this paper must show incontrovertible evidence of a slow, diabatic intrusion to the stratosphere that is as definite as the already published material on the Australia Black Summer pyroCb event. The manuscript does not meet that challenge, in my assessment. The ANY event still calls for explorations such as this. But the complexity of the atmospheric situation is not resolved or clarified by F23's analysis to my reading. Absent that outcome, I cannot recommend this publication. Major changes are called for.

Below, we explain why atmospheric stability and cloud formation does not prevent diabatic transport of smoke from the troposphere to the stratosphere. Nevertheless, we do not claim that diabatic heating is the only possible explanation. It is clear in our manuscript that we argue that diabatic heating is a likely explanation for the addition of smoke more than one week after the PyroCb formations. This is for example evident in Section 3.8 (on Line 345 in the track changes version of the revised manuscript) where we write: *"…The increasing potential temperature over time indicates that they were subject to self-lofting by radiation heating…".*

**Writing that we see indications of self-lofting shall not be seen as** *"…extraordinary claims…".* **It is rather the opposite.** We present evidence to support this. For example, in Figure 8 (and Figure 11), where it is evident that additional smoke entered the stratosphere more than one week later than the PyroCb formations in January.

The reviewer points to previous estimates of smoke by Peterson et al. (2021). We argue that these data are less trustworthy than our CALIOP data since the method used in Peterson et al. leads to misclassification of tropospheric smoke as being stratospheric. Please see our discussions on this below.

We have performed extensive data analysis with a high-resolution aerosol dataset from CALIOP. CALIOP is the only satellite sensor that can retrieve the smoke layers' position relative to the TP and simultaneously quantify the amount of smoke. Please see discussions on this below.

**General Concerns**

Regarding point 1 above, F23's messaging is that their data and interpretation were supported by Peterson et al. (2021). But my assessment is that the data are still contradictory and the Peterson et al. results are mischaracterized. F23 downplay the January-phase stratospheric injection by citing Peterson et al.'s revelation that the December-phase injection mass was 2-8 times greater than that of the December phase. However, the January-phase injection mass was by itself on par with the 2017 Pacific Northwest pyroCb event (PNE), which was at that time unprecedented in the satellite record for stratospheric smoke perturbation. Hence it is reasonable to imagine an Australia event that consisted only of the January phase. The Peterson et al. stratospheric smoke-mass injection could logically have resulted in a lasting plume on par with PNE. Peterson et al. calculated the stratospheric smoke source term by two methods, one based solely on the UVAI. The UVAI maximum, combined with the stratospheric area of the UVAI plume, showed that the January-phase event was on par not only with PNE, but all other historical documented pyroCb events as well.

We disagree. Our results are mostly in line with those presented in Peterson et al. However, we note that our results differ in the analysis of smoke from the January 4 fires. The tropopause altitude presented in their figure of CALIOP data of smoke from the Jan 4 fires (their Figure 8) are several kilometers lower than the TP included in the CALIOP files provided by NASA. The authors refer to TP data from radiosoundings at the Wagga Wagga station. We checked the temperature radiosonde data from Wagga Wagga (see Figure 1 below) and find TP altitude of >16 km @Jan 4 and >15.5 km @Jan 5. This is 3.5-4 km higher than what Peterson et al. used when estimating the amount of smoke injected to the stratosphere by PyroCbs. Hence, upper tropospheric smoke was misclassified as stratospheric, resulting in overestimation of the smoke mass.

Furthermore, their mass estimates are largely based on UVAI. Peterson et al. used a UVAI value of 15 as marker for stratospheric smoke. This value is not a hard limit between tropospheric and stratospheric smoke, which is evident in their figures (Figure 8 in Peterson et al.) where mid-tropospheric smoke (at 3.5-6 km altitude) is apparently misclassified as stratospheric (UVAI >15).

[Figure]

Figure 1. Temperature soundings at the Wagga Wagga station, southeast Australia, on January 4 and 5, 2020.

The UVAI does not contain high-resolution vertical information, limiting the possibility of distinguishing its altitude relative to the TP. The UVAI values are dependent on not only altitude, but also on the particle properties, concentration, and mass. High UVAI values can therefore appear also in the mid or upper troposphere leading to misclassification of tropospheric data as stratospheric. Hence, high UVAI values cannot be viewed as hard evidence for stratospheric smoke intrusions.

In summary, the UVAI based approach used in Peterson et al. holds uncertainties and possible errors affecting the interpretation of the event. CALIOP's altitude information is reliable.

As useful and advantageous as CALIOP data are for the purpose of characterizing nascent pyroCb plumes, they are—like all other remote sensing data items--subject to the vagueries of optimal sampling. The nature of sampling and the naturally small footprint of a nascent smoke plume is that they are often missed or incompletely captured with satellite data like CALIOP. My intense experience with the study of the ANY event, no doubt the same as F23's, revealed that the January-phase smoke plume was missed by CALIOP such that ideal plume height and concentration was delayed for several days. There were missing complete orbits or orbit segments on some days, and other days where CALIOP was just unlucky. Our survey of CALIOP showed that CALIOP made some luckier samples of the nascent December plume, as shown in F23's figures. This goes to the heart of why I asked in the first review about daytime CALIOP curtains and OMPS-LP, even conceding the poorer quality of those data. F23 showed in their supplement several daytime CALIOP curtains, implying the recognition of their value toward a more complete characterization of the nascent smoke plume. The unfortunate lack of perfect CALIOP sampling should not be convolved in the interpretation of plume altitude and concentration. In the case of ANY and PNE, the UVAI-based estimates of the immediate stratospheric presence of smoke are an independent and robust marker

of the stratospheric smoke source term. In short, there is a large, built-in uncertainty to the early sampling of CALIOP, even if all the CALIOP data are utilized. Unless F23 contend that the Peterson et al. January-phase stratospheric smoke mass is largely overestimated, the Peterson et al. results stand as an indication that the 4 January pyroCb event impregnated the stratosphere on par with almost all prior pyroCb events. If the veracity of the Peterson et al. estimates are not disputed, the challenge of showing additional smoke entering the stratosphere thereafter is considerable. F23 are encouraged to consider this argument and make whatever changes are called for in their line of analysis.

We have made additional analysis including the daytime data after the Jan 4 event. This analysis fully support the results in the nighttime data. Only very minor fractions of the smoke clouds from the Jan 4 fires were directly injected into the stratosphere. Together the daytime and nighttime data provide good coverage of the smoke clouds. The movement of the smoke clouds and the CALIOP orbits ensure that different parts of the smoke clouds are surveyed on different days. It is highly unlikely that a large stratospheric injection of smoke aerosol from the Jan 4 event would be completely missed by CALIOP. Our conclusion is that only small amounts of smoke entered the stratosphere via PyroCbs. Most of the smoke from the 2nd event entered the stratosphere later. This is shown in the supplementary, where we have added daytime curtain plots, and in Figures 8 and 11 (the first ~10 days after the Jan 4 fires). The daytime curtain plots are not used to estimate the height of the individual smoke clouds since these images contain high levels of noise, in particular in the depolarizing ratio. We have also added two adjacent nighttime curtain plots not previously included in the supplementary and estimated the height for this smoke layer in these plots (Figs S33 d-g). This has resulted in an additional data point in Figure 11.

CALIOP is the only reliable satellite instrument available that can distinguish both the smoke particle's position relative to the TP and quantify the smoke AOD. Other satellite instruments with vertical resolution become saturated. Finding the top of smoke layers is not the same as quantifying the smoke impact on the stratospheric aerosol load. This is true for all sensors.

- Regarding OMPS-LP, it cannot acquire quantitative data for dense aerosol layers. It is not reliable until more than a month after the PyroCbs during the Australian fires (Dec 29, and Jan 4).
- Regarding the UVAI, it cannot acquire vertical resolution data. One can only make assumptions on UVAI values (as described above). Hence, it cannot be used as hard evidence for a smoke layer's position relative to the TP.
- Regarding the mass estimates in Peterson et al., the TP altitudes are lower in their Figure 8 than those provided by NASA and lower than the radiosounding data as described above. This must have resulted in higher mass estimates in Peterson et al. (2021). Furthermore, their Figure 8 illustrates an example of smoke from the Jan 4 fires. It shows UVAI values >15 for tropospheric smoke, i.e. misclassifying tropospheric smoke as stratospheric. Using the tropopause altitude in NASAs curtain plots leads to almost no stratospheric impact at all. This is true for all CALIOP data. We see little direct stratospheric impact after the Jan 4 fires.
- Regarding cloud top temperatures, Peterson et al. show brightness temperatures (BT), of which the lowest BTs coincide with those in the upper troposphere in the CALIOP curtain plots shown in our supplementary file. BTs in the TP region do not tell the exact cloud top positions relative to the TP due to the minima in the temperature profile in the TP region (inversion in the stratosphere).

We have performed extensive analysis on the fire events' impact on the stratosphere and studied the time duration of stratospheric impact. We have evidence in the particle depolarization ratios that large smoke layers enter the stratosphere more than one week after the PyroCb formations at

January 4 (Figure 7 and 8). We see that the stratospheric scattering ratio, extinction coefficient, and particle depolarization ratio all increase more than one week after the January 4 fires. By investigating individual smoke layers, we find evidence that stratospheric smoke layers rise into the stratosphere from the troposphere (Figure 11). It is evident in the smoke layers' tops, mid-points, and bases.

On the contrary, no one has shown that all stratospheric smoke impact from the Jan 4 fires were a result of PyroCbs. The more we investigate the data, the less probable that scenario looks.

On the second point above, F23 made it much clearer in this revision how they accounted for the separation of December and January event smoke layers. Their CALIOP survey in Figures 1, 2, and Supplementary was more complete; their connection to phase 1, phase 2, and "other" was reasonable. From those data they built the central figures 7 and 11. This made it easier to understand these two important figures. But in my assessment, it did not adequately prove a diabatic pathway from the troposphere to the stratosphere, which is their central claim. Figure 7 contains only stratospheric observations. The CALIOP data assigned either to December or January are wholly or largely tied to two isolated, contained, and circulating smoke plugs followed by Allen et al. (2020 https://doi.org/10.1175/JAS-D-20-0131.1), Kablick et al. (2020), Khaykin et al. (2020), and Schwartz et al. (2020, https://doi.org/10.1029/2020GL090831). F23 expressly tied their December-phase data to one of these "isolated" entities. They did not directly tie their January-phase CALIOP data to an isolated smoke entity, but Figure 7a maps out the path of the plume in a way that compared almost perfectly with that of a smoke-vortex element illustrated in the above-cited papers. The important point is that the morphology of both of these smoke entities is wholly achievable by quasi-Lagrangian means due to their confinement and the absence of compromising uncertainties regarding smoke decay. These two contained smoke vortices perceptibly rose diabatically, in comparison to the extra-vortex ANY plumes (as hinted at by F23's "other" CALIOP data points in Figure 7).

We disagree. Our central claim is that smoke was transported from the troposphere to the stratosphere more than one week later than the PyroCb formations on January 4. We write that our data indicate diabatic transport (see for example Section 3.8 where we write: *"…The increasing potential temperature over time indicates that they were subject to self-lofting by radiation heating…"*).

F23's expressed and incidental following of these contained smoke vortices stands in contrast to the tropospheric CALIOP data they present. No attempt was expressed to follow these CALIOP layers materially. F23 clearly outline the difficulty of characterizing tropospheric smoke morphology because of the various, dominant, ubiquitous processes such as wet deposition. Canonical application of these forces to tropospheric smoke would lead to a vertically stratified decay profile that could appear as identical to the evolution of the CALIOP tropospheric smoke observations in Figure 11. For this reason alone, the apparent slope upward in the troposphere in Figure 11 has more than one explanation, unlike the same slopes shown for the stratospheric subset.

We can think of two atmospheric characteristics: atmospheric stability and cloud formation. The first point is less restrictive in the troposphere than in the stratosphere and should thus facilitate self-lofting compared with the conditions in the stratosphere. Cloud formation is mentioned in one passage by the reviewer:

*"…The fact that all of the ANY contained plume elements shown by Schwartz et al. were defined by water vapor enhancement, and that they all were based in the lowermost stratosphere, indicates a commonality, that being a direct pathway via pyroconvection. If that is a defensible statement, the*

*challenge for F23 is to show that slowly ascending air masses in the troposphere can deliver not only smoke through the cold-point tropopause, but water vapor plumes as well…"*

The last sentence seems to suggest that the vertical wind-speed must be high for the smoke to pass the cold point tropopause. Two of the three authors of this paper are experienced cloud scientists with several articles on clouds to their names (Martinsson (in the 1990ies) and Sporre (from 2010 onwards)), and we disagree. Martinsson et al (2022) examined the water content of the PNE smoke layers. We (Martinsson et al 2022) found $H_2O$ concentrations of $7 - 14$ ppmv in the smoke layers at atmospheric pressure levels lower than 110 hPa, implying a maximum $H_2O$ vapor pressure of 0.16 Pa corresponding to a few percent relative humidity (RH). If we move this dry air to the ground the $H_2O$ concentration will increase by about a factor of 10. At the same time the temperature increases. A typical ground temperature in the summer is at least 15° C. Comparing the compressed $H_2O$ vapor pressure with the saturation vapor pressure at that temperature we find a RH of the order of 0.1%, This implies that almost all the water was precipitated before the PNE pyroCbs reached the stratosphere, e.g., if the air originally held 50% RH at the ground, 99.8% of the water was precipitated on the way to the stratosphere. That did not prevent large amounts of smoke from reaching the stratosphere. The water content of the December ANYSO was within a factor of 2, implying that almost all the water was lost on the way to the stratosphere also in this case. Smoke that reaches the upper troposphere by pyroCb transport has experienced similar air mass history because the upper troposphere is almost as cold as the cold point tropopause.

Would then a slow passage of the cold point tropopause make a drastic difference compared with a fast, convective passage? Fast vertical transport means fast cooling rate and thus high production rate of condensable water. Because of a lag in the condensation rate, the maximum supersaturation of the cloud will be high compared with slowly rising air with the same aerosol content. As a result, more cloud particles can form in fast vertical transport and take part in the later formation of precipitation. The maximum supersaturation is also strongly dependent on the amount of aerosol present. Dense aerosols, like wildfire smoke, strongly limit the maximum supersaturation of a cloud because of the increased size of the water sink and hence the condensation rate becomes much higher with a low maximum supersaturation as the result. Therefore, a larger fraction of the aerosol remains as unactivated interstitial aerosol particles. The reason that wildfire smoke reaches the stratosphere in such large amounts is the extremely high aerosol concentration, whereas a "normal" Cb loses almost all its aerosol mass because of the high production rate of condensable water in the high updraught velocities is not curbed by a large water sink, making a large fraction of the aerosol available for cloud particle formation. In conclusion, the dense smoke lifted by the pyroCbs to the upper troposphere has already withstood losses of 99% of the water or so due to precipitation. The final percent or permilles of water cannot precipitate all the remaining dense smoke, only the aerosol particles with most affinity for cloud particle formation, regardless if the transport is fast or slow.

In conclusion, smoke that has reached the upper troposphere has already lost most of the water to precipitation. A large fraction of the aerosol escapes precipitation because the smoke is dense.

The difficulty of analyzing tropospheric data lies in the frequent presence of clouds. Aerosol layers below the smoke layers complicate the attenuation correction used to compute the AOD.

Regarding the slope in Figure 11, there are no indications of large immediate stratospheric impact in the entire set of CALIOP swaths. As mentioned above, CALIOP is the only satellite sensor that can adequately distinguish smoke layers' position relative to the TP.

The supplement plots by themselves cannot be used to infer upward transport. Static images 24 hours apart allow no such definitive statement. Moreover, I could not discern a systematic rise in the

tropospheric smoke from these figures. If indeed there is such a signal, it would be essential for F23 to analyze and defend that scenario.

We have now included daytime data increasing the time resolution to 12 hours. CALIOP sampled the smoke multiple times each day. None of these curtain plots show, indicate, or even suggest any large direct stratospheric impact of smoke from the January 4 fires. Figure 11 is a direct compilation of the CALIOP curtain plots presented in the supplementary file. The figure does in itself provide this information.

F23 should be expected to offer other potential explanations for the pattern seen in FIgure 11. Given that wide-scale tropospheric smoke pollution on par with that seen in this case is not uncommon, and other such occurrences were not followed by significant stratospheric pollution, alternate explanations should be given at least equal weight to diabatic lofting.

We disagree. This is not a random event with faint smoke layers that dissipates quickly. The horizontal extensions of these layers are 100s to 1000s of kilometers, and PyroCbs injected them to high tropospheric altitudes. It is not surprising that these dense large smoke layers are transported differently than smaller wild-fire injections into the troposphere.

It is noteworthy that the smoke vortex associated with F23's January-phase CALIOP data was tracked by Schwartz et al. by virtue of MLS stratospheric water vapor enhancements. If this smoke entity had its origin in the troposphere, it would have been subject to the cold trap at the tropopause, a limiting factor on its water vapor content. The fact that all of the ANY contained plume elements shown by Schwartz et al. were defined by water vapor enhancement, and that they all were based in the lowermost stratosphere, indicates a commonality, that being a direct pathway via pyroconvection. If that is a defensible statement, the challenge for F23 is to show that slowly ascending air masses in the troposphere can deliver not only smoke through the cold-point tropopause, but water vapor plumes as well.

Please see our answers above.

Figure 8 is composed of zonal averages. The range from 20-80S would embody tropopause heights generally above 14 km in the northern realm to lower in the southern realm. Since this analysis is done with an absolute altitude scale, it is to be expected that the data below 14 km is a blend of tropospheric and stratospheric aerosol. Since these data almost certainly represent some unknown blend of stratospheric air, it is uncertain as to how to assess these lowermost data points in relation to those above 14 km.

Tropospheric data are not included in the figure. The figure shows stratospheric air only (as indicated in the figure caption). Data below 14 km was not used for the analysis in Figure 9.

This figure shows that additional smoke entered the stratosphere more than one week after the Jan 4 fires. There is a large significant difference in particle depolarization ratio between smoke from the December and January fires. This shall not be neglected. Figure 8 is hard evidence. There is no sign of large immediate stratospheric impact. Instead, we observe addition of smoke in the stratosphere more than one week after the Jan 4 fires.

Both phases produced UTLS smoke with equally small depolarization ratio (depol for short). E.g., see the intense stratospheric layer in Figure S3b,c. The fact that the December phase produced such low depols at such high altitude prompts the question as to the true difference between the December and January nascent UTLS plumes' particle-shape populations. Could F23 comment on that?

This CALIOP curtain is the only indication of such low depolarization ratio from smoke from the 1st fire event. We do not know why it is lower than in the remaining CALIOP curtain plots.

There is a clear separation in smoke depolarization ratios between the two fires. The particle depolarization ratios for smoke from the 1st fire increased before smoke from the 2nd event entered the stratosphere. This is shown in Figure 7 and 8.

F23 have convincingly shown that the January-phase stratospheric depol is somewhat less than the December phase. But what does that necessarily say about its origin altitude? Wouldn't we expect the photolytic process to drive both depol populations to the same eventual value? If so, what would account for the difference in the December and January aged plumes at stratospheric altitudes?

It is well known that depolarization ratios are lower for tropospheric than stratospheric smoke. A likely explanation for this is that the soot agglomerates collapse when smoke particles are exposed to water. Smoke particles from the December fires are therefore expected to be more irregularly shaped than smoke particles from the January fires. The fact that the particle depolarization for smoke from the January fires is so much lower than for smoke from the December fires is a strong indication that the smoke was processed in the troposphere before entering the stratosphere. The particle depolarization ratio may not necessarily become equal for smoke from these two fires. Smoke particles with collapsed soot agglomerates (January 4 fires) may not reach as high final particle depolarization ratios as smoke that were not processed in the troposphere before entering the stratosphere.

Peterson et al. showed that, by number, there were many more pyroCbs in the December phase that injected only to the troposphere than on 4 January. That may or may not have been evident in the CALIOP curtains shown in F23, but it is plainly evident in the full set of CALIOP curtains that there was abundant tropospheric smoke from 29 December onward. Given Peterson et al's accounting, doesn't it seem reasonable that there was much tropospheric smoke from the December phase that was in place by the time of the January phase?

We separated data using a combination of the UVAI, backscattering coefficient, and depolarization ratio. This is a reliable method to distinguish the separation of smoke layers in time and space (both vertically and horizontally). Tropospheric smoke from the December and January fires were separated in time and space enabling the classification of smoke.

Technically, F23 do not show diabatic transport from below the tropopause. They infer it. The stratospheric fraction of observations can be accepted as reflecting diabatic lofting largely on the strength of prior publications such as Khaykin et al;, Kablick et al. and Schwartz et al., as well as the several papers on the PNE pyroCb event. But the troposphere-to-stratosphere mechanism has still not been proven with observations.

It is unclear for us why the reviewer claims that transport of smoke from pyroCbs terminating in the upper troposphere to the stratosphere is unexpected, see explanations above.

We present evidence for additional cross-TP transport of smoke occurring more than one week after the PyroCb formations. The data indicate diabatic lofting from the troposphere. Such transport was neglected until recent studies.

No study has shown that PyroCb formation is the only transport path to the stratosphere for wildfire smoke. In a recent study Ohneiser et al. (2023) investigated the potential for self-lofting from the troposphere to the stratosphere. They find that dense smoke layers can rise from the troposphere

via self-lofting. Our data indicate that diabatic heating transported smoke from the troposphere to the stratosphere.

In terms of the flow of argumentation in F23, I found several instances where they assert their conclusion about the diabatic lofting from the troposphere prior to any detailed analysis. In this way they seemed to put the cart before the horse. It is advised that they not only bolster their analysis proving the tropospheric diabatic lofting, but also withhold any conclusions/assertions until the reader sees the proof.

We did not present self-lofting before showing the evidence for it. In the revised manuscripts (submitted in June and August), self-lofting is mentioned in the Introductions section together with citations to previous studies. Next time it is mentioned is at the end of the Discussions section (section 3.8 Smoke transport into the stratosphere). Note that all 11 figures were presented and discussed before we mentioned Self-lofting in the discussions section.

F23, on occasion, refer to the smoke-layer tops in the LMS as "minor" in relation to the bulk of the aerosol layer. This is reasonable from a descriptive perspective, but considering the strong vertical stratification of aerosol lifetime, this may be prejudicial. There is no doubt that typical pyroCb events distribute smoke throughout the troposphere up to the UTLS, leaving what may appear to be a minor portion at the topmost altitudes. But that "minor" part has a much greater potential to last than the eye-catching tropospheric parts. Moreover, sampling by CALIOP may give the strong appearance of the topmost smoke as being small in proportion to lower plumes. A telling example of this is shown in Figures 6 and 7 here: **https://doi.org/10.1029/2021JD034928.** It shows an early view of the PNE smoke plume that captured a small footprint of the pyroCb smoke plume. This "minor" feature represented the most consequential early indication of the smoke that polluted the stratosphere. Hence F23 are asked to reflect on the "minor" indications of the January-phase LMS smoke as early as 6 January illustrated in Figure 11.

It is clear from the CALIOP curtains that large amounts of smoke enter the stratosphere more than one week after the PyroCb event Jan 4. This was not the case for the 2017 North American fires, where dense smoke layers were observed up to two kilometers into the stratosphere on the second (Aug 14) and third day (Aug 15) after the fire.

CALIOP obits the Earth 14.5 times per day resulting in 29 times of sampling per 24 hours (14.5 night + 14.5 day data). It evidently passed over the smoke layers on multiple occasions each day. This is shown in the supplementary file. The curtain plot on Jan 6, which the reviewer refers to, shows a faint smoke layer with backscattering coefficients more than one order of magnitude lower than the smoke layers from the North American fires in the study that the reviewer refers to (see Figure 2 below). The CALIOP data and curtain plots show that the smoke had little direct impact on the stratosphere.

[Figure]

**Figure 2. Illustration of smoke layers taken two days after PyroCb formation from the North American fires (left), and from the Australian January 4 fires (right). Arrows mark the dense smoke layers above the TP from the NA fires (left), and a faint smoke layer at the TP from the Jan 4 fires (right).**

F23 responded to my initial review's question about exploiting CALIOP color ratio. They argued that the differential attenuation made that an insurmountable hurdle. However, they did exploit CALIOP color ratio in Martinsson et al. (2022) in a manner I was alluding to. Hence, I still wonder if one could discern systematics in the temporal evolution of the color ratio. Both the color ratio and the differential attenuation are associated with particle size. There might be qualitative as well as quantitative tactics to assess the temporal changes even while acknowledging the attenuation issue. Since this group met that challenge in Martinsson et al., they are encouraged to do the same here or explicitly address the reason why they avoid that in this work.

The 1064 nm signal is tricky to use because of high noise. The weak signal from air molecules at 1064 nm makes the method we use to estimate the lidar ratio of 532 nm difficult (impossible) to use for the longer wavelength. We tried a different method this time compared with Martinsson et al (2022) but failed, probably due to the noise problem. Rather than going back to all files and changing to the method we used in Martinsson et al (2022) we opted to skip the color ratios this time.

Katich et al. (2022, DOI: 10.1126/science.add3101) used in situ aircraft data to develop a "fingerprint" of pyroCb stratospheric smoke, in comparison to non-pyroCb smoke, in terms of an especially large coating thickness around the BC core. F23 may wish to review this paper and comment on any potential conflicts with their hypothesis of photolytic processes reducing OA material mass.

There is no conflict with our results and Katich et al (2022). Their paper indicates that smoke particle sizes did not change from 2 months (Oct 2017) to 9 months (May 2018) after the NA fires. In Martinsson et al. (2022) we reported a rapid increase in the particle depolarization ratio during the first month. Also in the present paper we see a rapid increase in particle depolarization ratio during the first month, after which the particle depolarization ratio keeps a constant value. Hence, organics are depleted during the first month. Remaining particulate matter, i.e., soot and residual organics remain in the stratosphere until transported to the troposphere or depleted by other processes.

**Targeted Comments/Concerns**

Below, line numbers, figure numbers, and F23 quotes are in **bold**. My reaction is in plain text.

**L30:** The vertical definition of the LMS is not given herein. The term is used sometimes qualitativley, but also used as part of a targeted calculation of AOD. The details of that calculation need to be provided.

We added this information to Section 2.1, where we describe how the AODs where computed.

**L53-54, "At least 38 PyroCbs injected smoke to the stratosphere during two events…":** Please revisit Peterson et al. and revise this statement. According to Peterson, only a subset of the 38 pyroCb-pulse injections reached the stratosphere.

Thank you for pointing this out. We have update the text in our introductions section, i.e. 20 PyroCbs entered the stratosphere according to Peterson et al.

**L59-62:** Even with F23's revisions, the statements in this paragraph conflate two previous published conclusions under the banner of transport from the troposphere to the stratosphere. Only Ohneiser et al. fits within this pathway. The Peterson and Khaykin papers start the plume ascent in the lowermost stratosphere. Please add the proper nuance.

We agree that Ohneiser et al. is the paper that describes this. This is well expressed in our manuscript in the second sentence on **L59-63**, where we write: *"…Most smoke encounters in the stratosphere have been explained through upward transport by pyrocumulonimbus clouds, but studies in recent years suggest that further transport mechanisms cause cross-tropopause transport of smoke. The North American wildfires in Aug 2017 showed that self-lofting by radiative heating of the dense smoke layers caused smoke to rise from the tropopause into the LMS (e.g. Khaykin et al., 2018; Peterson et al., 2018). Ohneiser et al. (2021) suggested self-lofting of smoke from the mid-troposphere as cause of extensive aerosol layers in the Arctic stratosphere in the end of 2019 and beginning of 2020.…".*

**L73, "…aerosol stayed in the stratosphere for a year…(Ohneiser et al. (2022).":** As far as I can tell from Ohneiser (2022), they only show PNE smoke for ~8 months. Did I miss it? If not, please provide another citation.

We agree. Thank you for pointing this out. We have changed this in the manuscript.

**L125-126, "The depolarization ratios for smoke from the 2nd fire were clearly lower than those for smoke from the 1st fire…":** While some systematic difference is visually apparent, there is a considerable overlap in depol between the two phases. The December phase generated some very low-depol layers above the tropopause, as did the January phase. There are several additional CALIOP curtains attributable to the December phase, not shown here, that reinforce the realization of overlap in the depol ratio between phases. In general, it appears the free tropospheric smoke depol has single-digit depol, stratospheric has decidedly double-digit depol., and tropopause-level smoke has a wide range, as manifested in both phases. Would the authors care to comment on that?

We agree that the depolarization ratio is lower for tropospheric smoke, but there is a clear significant difference in the particle depolarization ratio values for the December and January smoke in the stratosphere.

**L125 paragraph and figures called out**: It would be helpful to have marks such as arrows on the figures pointing to features the authors want to highlight to make their point.

We do not wish to make the figures more busy. The differences in depolarization ratio is clear with or without arrows.

**L127:** What property? F23 describe two populations of smoke depol, but not in contrast to other particle types. Please elaborate.

We have changed the sentence to *"…This difference remains for more than one month, i.e., smoke layers from the 2nd fire continues to have lower depolarization ratios than smoke from the 1st fire…"*

**L231-232, "Peterson et al. (2021) reported much larger stratospheric impact from the 1st fire, based on studies of the fires' immediate impact.":** Yes, but the stratospheric mass from Phase 2 was equivalent to PNE, according to Peterson's Figure 1. So, on its own merits, the Phase 2 plume was a major stratospheric presence.

We find that the methods used in Peterson et al. misclassifies tropospheric smoke as stratospheric. Please see answer above. Hence, the mass estimates in Peterson cannot be used as a baseline for comparison.

**L197 and elsewhere:** "elevation" is regularly used to characterize an increase in AOD. This term also denotes changes in altitude. It might be advisable to choose another descriptor of the AOD amplitude change.

We agree that this could be confusing and have made changes in the manuscript using the word "increase", instead of elevation.

**L232-233, "10 days after the PyroCb formations we start to see more stratospheric influence (Figure. 7).":** Figure 7's first January-phase data point is on 14 January, ten days post event. But Figure 2 and especially supplementary figures show January-phase CALIOP curtains dating to 5 January (1 day post event). Moreover, Figure 11 starts on 5 January. But the reader is first introduced to the January-phase smoke by the callout to Figure 7. So, it seems to be misleading to support the above statement by this figure callout. Figure 11 shows stratospheric influence from the January phase being first detected by CALIOP on or about 6 January. The weight of Figure 2, 11, and the supplementary figures indicates that a re-characterization of this sentence is called for.

With the words *"…we **start to see more** stratospheric influence…"* we point to increasing smoke abundance in the stratosphere.

**L233:** It's not clear what **"stratospheric influence"** means. Figure 7 simply follows two smoke vortices. This doesn't represent the entirety of the smoke plume. Please consider rephrasing this.

Stratospheric influence means that the smoke influences/impacts the stratosphere.

**L236, "Over time, more and more smoke…":** This is not obvious from the CALIOP curtains in FIg 1,2, and Supplement. This is a conclusion stated before any proof is given.

We have changed the sentence to *"Over time, more and more smoke appears in the stratosphere"*.

**L239, "…rose by at approximately the same rate as…":** Again, Figure 7 simply follows two vortices, one of which was spawned by the January plume. So it is no surprise; the ascent of that vortex has already been documented by Khaylin, Kablick, and Schwartz.

We too see ascension. We do not claim that this our major finding in the manuscript.

**L257-258, "Some of the smoke from the 1st event reached the UT (Figure. 1) and may have risen later along with smoke from the 2nd event contributing somewhat to the second AOD peak.":** This is putting the cart before the horse. The reader has yet to see any analysis proving tropospheric lofting.

We have deleted this sentence in the updated manuscript.

**L259, "Smoke from the 1st event rose markedly in the stratosphere before smoke from the 2nd event entered the stratosphere (Figure 7b)."**: This conclusion cannot be drawn from Fig 7. F23 started the plots on 14 January for the January plume. The various CALIOP curtains prove that there was stratospheric smoke from the Jan phase many days earlier. See also Peterson et al.

Figure 8 (and 7b) shows that additional smoke appeared in the stratosphere more than one week after the PyroCb formation, Jan 4. See also our comments with regards to the Peterson et al paper above. We added a reference to Figure 8.

**L265, "…the 2nd event that ascended later into the stratosphere."**: At this point the reader has not been shown proof that tropospheric smoke ascended into the stratosphere. The material that has been presented at this point does not perform that task.

We disagree. Figure 8 shows that smoke appeared more than one week after the PyroCb formation (Jan 4).

**L295-297, "This explains the low depolarization ratios for smoke from the 2nd event…"**: See my comments about the copious observations of low-depol. tropospheric smoke in the December plume. Note also the multiple CALIOP observations of low-depol, tropopause-level smoke from both events.

There is a significant difference in the particle depolarization ratios for smoke from the 1st and 2nd event fires.

**L326-327, "From the 1st event we do not see evidence of extensive crosstropopause transport beyond the initial PyroCb…":** The reader does not get any information aligned with this conclusion. What analysis did F23 perform to elicit this finding?

It is evident in the supplementary file as well as in Figure 8. We added a reference to these to the sentence.

**L354-355, "A continuous crosstropopause transport over the course of several weeks also affects the AOD evolution."**: This pathway is taken as a given here. Cross-tropopause transport is not quantified in a manner to support the claim that it occurred over several weeks.

We changed this sentence to *"…The AOD evolution (Figure 9a) suggests cross-tropopause transport over the course of several weeks…"*

**L356-357, "Our study indicates that smoke from the 2nd event had larger long-term impact…(Figure 9."**: Nowhere, to my reading, did F23 explain how Figure 9a was constructed. Elsewhere in the paper they described tracking the two phases of smoke only out to 4 February. How did the smoke-phase distinction over several months get accomplished?

Figure 9a and b is based on stratospheric zonal means as stated in the figure legend. Tracking the smoke until February refers to analysis of individual smoke layers. Regarding how the figure was constructed, it is described in the manuscript a few lines above (Section 3.5): *"…We use this minimum to separate smoke data from the two fires (dashed lines in Figure 8) to form the AOD of the two events and investigate their individual impact on the stratospheric AOD…"*

We changed the figure caption to *"…Smoke decay in the stratosphere. a) 8-day running mean of the background subtracted stratospheric zonal mean AOD at 20-80°S above 14 km altitude for the 1st and 2nd event, respectively. b) Daily means of background subtracted AODs for the 1st event only, and c) smoke data from individual smoke layers from the dense isolated smoke from the 1st event*

*(scattering ratios, SR, from CALIOP) normalized with water vapor concentrations (cH2O, from MLS). The exponential fits correspond to a smoke half-life of 10 ± 2 (b) and 10 ± 3 days (c)…"*

**Figure 5**: Do F23 wish to opine on the secondary AOD increase in the winter months? This period is not a focal point of the paper but the feature of increased AOD is in stark contrast to the decay signal, thus potentially more consequential than the earlier AOD dip and rise on which F23 focus.

In periods with stratospheric background (no smoke or volcanic aerosol) the stratospheric AOD varies with season due to air transport within and out of the stratosphere, and due to varying TP height (the extratropical stratosphere contains more air in the winter. This is well-known and well documented in literature, and have little impact on the shape of the stratospheric AOD after the fires.

**Figure 7 caption, "…smoke transport and chemical evolution…":** "chemical" is not shown, only optical properties. Chemistry is inferred from simulations, which is discussed in the text. But the figure caption should describe what is shown.

We agree. We have changed this in the manuscript and write "smoke particle evolution" instead of "chemical evolution".

**Figure 9 caption, "Smoke decay in the dense isolated cloud from the 1st event..":** This first sentence is confusing. The figure panel a shows both events. And panel a's description is "zonal means." How are events segregated while calculating zonal means?

We agree, and changed to *"Smoke decay in the stratosphere"*.

**Figure 11.** The fit lines are not described in the caption.

We have added this information to the figure caption.

**Supplementary Material, "Some Caliop curtain plots were excluded from the analysis since the aerosol attenuation signals were too weak even though the UVAI indicate presence of smoke aerosol.":** This is unclear. Since there is no one-to-one relation between night CALIOP and day UVAI, how did F23 determine the weakness threshold used for excluding CALIOP layers?

We realize that this text was unclear and have rewritten this section in the supplementary. It now reads: *"All CALIOP curtain plots with distinct aerosol layers were included in the analysis"*.

**Technical Comments**

**Figure S1:** Please separate the left and right columns. The continuous black background makes it difficult to see the east edge of the left column and west edge of the right column.

We agree that this is helpful and have made this change in the supplementary file.

Throughout the text, use subscript notation in "SO2."

We agree, and now use subscript notation.